# Retinal microvascular and neuronal pathologies probed in vivo by adaptive optical two-photon fluorescence microscopy

Qinrong Zhang[1,2†], Yuhan Yang[1†], Kevin J Cao[2,3], Wei Chen[1,2], Santosh Paidi[4], Chun-hong Xia[4,5], Richard H Kramer[2,3,5], Xiaohua Gong[4,5], Na Ji[1,2,3,5,6]*

[1]Department of Physics, University of California, Berkeley, United States; [2]Department of Molecular and Cell Biology, University of California, Berkeley, United States; [3]Helen Wills Neuroscience Institute, University of California, Berkeley, United States; [4]School of Optometry, University of California, Berkeley, United States; [5]Vision Science Program, University of California, Berkeley, United States; [6]Molecular Biophysics and Integrated Bioimaging Division, Lawrence Berkeley National Laboratory, Berkeley, United States

*For correspondence:
jina@berkeley.edu

†These authors contributed equally to this work

Competing interest: The authors declare that no competing interests exist.

**Abstract** The retina, behind the transparent optics of the eye, is the only neural tissue whose physiology and pathology can be non-invasively probed by optical microscopy. The aberrations intrinsic to the mouse eye, however, prevent high-resolution investigation of retinal structure and function in vivo. Optimizing the design of a two-photon fluorescence microscope (2PFM) and sample preparation procedure, we found that adaptive optics (AO), by measuring and correcting ocular aberrations, is essential for resolving putative synaptic structures and achieving three-dimensional cellular resolution in the mouse retina in vivo. Applying AO-2PFM to longitudinal retinal imaging in transgenic models of retinal pathology, we characterized microvascular lesions with sub-capillary details in a proliferative vascular retinopathy model, and found Lidocaine to effectively suppress retinal ganglion cell hyperactivity in a retinal degeneration model. Tracking structural and functional changes at high-resolution longitudinally, AO-2PFM enables microscopic investigations of retinal pathology and pharmacology for disease diagnosis and treatment in vivo.

## Editor's evaluation

The authors developed a two-photon fluorescence microscope coupled with adaptive optics (AO-2PFM), allowing in vivo imaging of the mouse retinal structure and function. This new imaging system will be important for exploring normal retinal physiology and pathological alterations in retinal disease models.

## Introduction

Retina is a layered tissue in the back of the eye that transduces light into electrochemical signals to be further processed by the brain for visual perception and cognition (*Kandel et al., 2021*). As one of the most energy-demanding tissues, the retina is metabolically sustained by an intricate vasculature with several laminar plexuses (*Selvam et al., 2018*). Vascular and neuronal abnormalities in the retina are associated with both ocular (*Schachat et al., 2017*) and systemic diseases (*Cheung et al.,*

*2017*; *London et al., 2013*; *Lechner et al., 2017*), underscoring the importance of studying retinal pathology and pharmacology.

With well-developed genetics and similar physiology to the human retina, mouse models have been widely utilized for mechanistical studies of retinal diseases. Behind highly transparent mouse eye optics (i.e. cornea and crystalline lens), the retina is uniquely accessible to light and the only part of the nervous system that can be probed non-invasively by optical imaging. Recent advances in mouse genetics have enabled fluorescence microscopy investigations of vasculature (*Ivanova et al., 2021*) as well as neurons and glial cells (*Jo et al., 2018*; *Martersteck et al., 2017*; *Eme-Scolan and Dando, 2020*) of the mouse retina. Among fluorescence microscopy techniques, two-photon fluorescence microscopy (2PFM) (*Denk et al., 1990*) utilizing near-infrared (NIR) excitation is particularly suited for retinal imaging. Its intrinsic optical sectioning capability permits depth-resolved three-dimensional (3D) imaging throughout the retina. With the retinal photoreceptors minimally responsive to NIR light, 2PFM is also an ideal tool for functional studies of retina (*Euler et al., 2002*; *Baden et al., 2016*). However, as a far-from-perfect imaging system, the optics of the mouse eye introduce severe aberrations to the NIR excitation light, preventing high-resolution visualization of subcellular features in vivo. As a result, the vast majority of microscopy studies have been carried out ex vivo on dissected retinas, preventing longitudinal investigations of retinal pathology under physiological conditions.

Adaptive optics (AO) is a collection of technologies that actively measure and correct for optical aberrations (*Hampson et al., 2021*), and has been applied to optical microscopy for high-resolution imaging of neural tissues (*Ji, 2017*; *Rodríguez and Ji, 2018*). It has also been combined with ophthalmological imaging modalities to restore diffraction-limited imaging performance for the human retina (*Liang et al., 1997*; *Akyol et al., 2021*). Because of the severe aberrations of the mouse eye, AO has also been applied to in vivo imaging of the mouse retina (*Palczewska et al., 2014*; *Biss et al., 2007*; *Alt et al., 2010*; *Geng et al., 2012*; *Sharma et al., 2013*; *Wahl et al., 2016*; *Wahl et al., 2019*; *Qin et al., 2020*). However, there are disagreements in the reported spatial resolutions (*Palczewska et al., 2014*; *Biss et al., 2007*; *Alt et al., 2010*; *Geng et al., 2012*; *Sharma et al., 2013*; *Wahl et al., 2016*; *Wahl et al., 2019*; *Qin et al., 2020*), characteristics and magnitude of aberration (*Palczewska et al., 2014*; *Biss et al., 2007*; *Alt et al., 2010*; *Wahl et al., 2016*; *Wahl et al., 2019*; *Qin et al., 2020*), and the effectiveness of AO (*Biss et al., 2007*; *Alt et al., 2010*; *Wahl et al., 2016*; *Wahl et al., 2019*; *Qin et al., 2020*). For example, whereas previous papers reported cellular resolution without AO, a recent AO-2PFM study (*Qin et al., 2020*) reported extremely large aberrations in the mouse eye and found AO to be required in order to resolve microvasculature and cell bodies in 2D in vivo. These discrepancies have led to uncertainty over the imaging performance achievable with conventional 2PFM and the necessity of AO for microvascular and cellular investigations of retinal physiology. Together with a lack of detailed imaging protocols, they have prevented the routine application of AO-2PFM to disease diagnosis and therapeutic intervention in the retina of mouse models of ocular, cerebral, and systemic diseases.

The aims of this work are to provide a resource for in vivo retinal imaging using 2PFM, by optimizing the design of a 2PFM for in vivo imaging of the mouse retina, characterizing mouse ocular aberration, developing a guideline for adaptive optical 2PFM (AO-2PFM) imaging, and demonstrating its applications to retinal pathology and pharmacology. Using a carefully engineered 2PFM and following an optimized sample preparation procedure, we were able to achieve two-dimensional (2D) cellular resolution imaging performance in the mouse retina without AO. For synaptic, subcellular, and three-dimensional (3D) cellular resolution imaging of the mouse retina, AO was essential in improving image brightness, contrast, and resolution. Testing the performance of AO-2PFM in various transgenic mouse lines, we proposed strategies to maximize its impact on image quality improvement. We extended the application of AO-2PFM to mouse retinal pathology and pharmacology by imaging the retinas of two transgenic models with proliferative vascular retinopathy and retinal degeneration, respectively. In our model of proliferative vascular retinopathy, AO enabled us to, for the first time, characterize retinal vascular lesions with sub-capillary details over multiple days and observe cell migration in vivo. In our model of retinal degeneration, AO allowed high-fidelity interrogation of pharmacologically modified hyperactivity of retinal ganglion cells (RGCs), indicating AO-2PFM as a promising tool evaluating retinal pharmacology in vivo. Together, by systematically optimizing and applying AO-2PFM to in vivo mouse retinal imaging, our work represents an important advancement in enabling high-resolution longitudinal studies of retinal pathology and pharmacology for disease diagnosis and treatment.

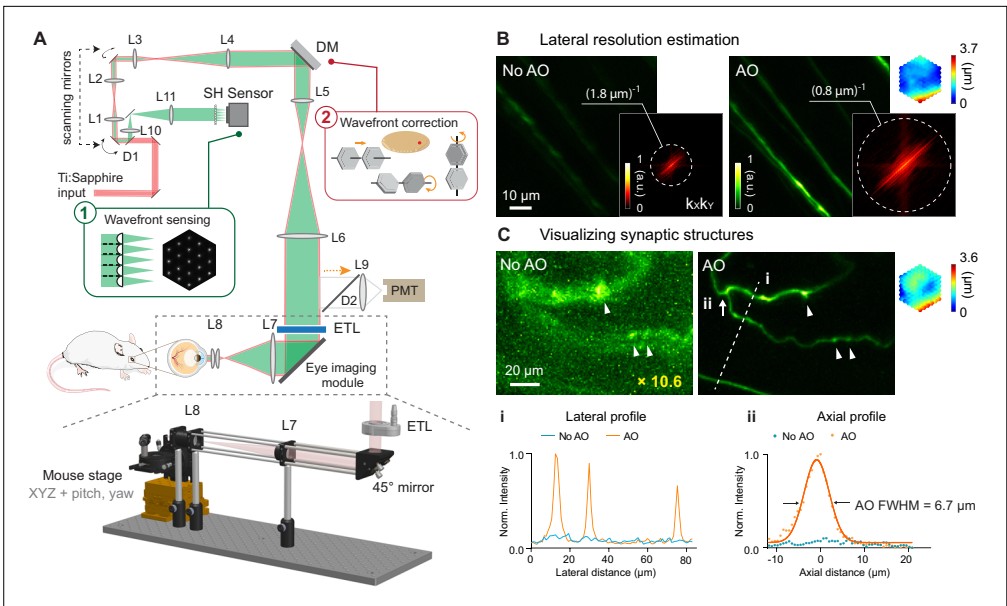

**Figure 1.** AO-2PFM for diffraction-limited imaging of the mouse retina in vivo. (**A**) Schematics of AO-2PFM. Inset 1: direct wavefront measurement by a Shack-Hartmann (SH) sensor composed of a lenslet array and a camera. Inset 2: wavefront correction with a deformable mirror composed of 163 segments with piston, tip, and tilt controls. Grey dashed box: eye imaging module. Bottom: 3D assembly of eye imaging module. L, lens; D, dichroic mirror; DM, deformable mirror; PMT, photomultiplier tube; ETL, electrically tunable lens. (**B**) Maximum intensity projections (MIPs) of image stacks (72×72×25 μm³) of RGC axons measured without and with AO, respectively, normalized to AO image. Insets: $k_X k_Y$ spatial frequency representation of the images and corrective wavefront. (**C**) MIPs of image stacks (132×97×32 μm³) of fine RGC processes measured without and with AO, respectively, normalized to AO image. 'No AO' image brightness artificially increased by 10.6× for better visualization. White arrowheads: putative synaptic structures. Inset: corrective wavefront. Bottom: **i**: lateral signal profiles along white dashed line i; **ii**: axial signal profiles of process ii (white arrow). Signals in the line profiles were normalized to the maximal value of the AO condition. Representative data from >3 experiments (technical replicates).

The online version of this article includes the following source data and figure supplement(s) for figure 1:

**Source data 1.** Source image stacks of retinal axons (*Figure 1B*).

**Source data 2.** Source image stacks of retinal neuronal processes (*Figure 1C*).

**Figure supplement 1.** Characterization of aberrations introduced by ETL and alignment.

**Figure supplement 2.** Contact lens and eye gel application improve image and wavefront sensing quality.

**Figure supplement 3.** Zernike decompositions and corrective wavefronts for all experiments.

# Results

## Optimized AO-2PFM for in vivo mouse retinal imaging

A home-built two-photon fluorescence microscope equipped with a segmented deformable mirror (DM) and a Shack-Hartmann (SH) sensor (*Li et al., 2020b*) was modified for in vivo mouse retinal imaging by replacing the objective lens with an add-on eye imaging module (*Qin et al., 2020*; *Grulkowski et al., 2018*; *Figure 1A*, Materials and methods). The module consisted of an electrically tunable lens (ETL) whose adaptive surface was conjugated to the DM, a turning mirror, and two lens groups (L7 and L8) that relayed the adaptive surface of the ETL to the pupil of the mouse eye. With this design, the optics of the mouse eye focused 920 nm light onto the retina to excite fluorescent markers and collected the emitted fluorescence for detection. The ETL allowed us to adjust the focal plane in the mouse eye without translating the mouse (*Jian et al., 2013*) or optics (*McNabb et al., 2019*) in the imaging system. For all experiments, system aberrations in the two-photon illumination path were measured with a modal AO method and corrected before image acquisition (Materials and methods; 'No AO' images: system aberration correction only).

To ensure optimal performance, we thoroughly characterized our AO-2PFM. We investigated how ETL current and mouse eye placement (with a longitudinal displacement of up to 4 mm in typical

experiments) impacted imaging performance (*Figure 1—figure supplement 1*). We found that aberrations introduced by the ETL at different control currents minimally affected image quality and that axial focal shift varied linearly with ETL current while field-of-view (FOV) size remained mostly constant. We also optimized sample preparation procedure. We discovered that a custom-designed 0-diopter contact lens (CL; design parameters in *Figure 1—figure supplement 2A*) in combination with a single application of eye gel between the CL and the cornea reduced aberrations, prevented cataract formation, and improved wavefront sensing and imaging for hours (*Figure 1—figure supplement 2*).

In order to achieve diffraction-limited imaging of the mouse retina in vivo, we measured and corrected ocular aberrations with a direct wavefront sensing method (*Wang et al., 2014b*; *Wang et al., 2015*), utilizing the SH sensor for wavefront measurement and the DM for wavefront correction (*Figure 1A*). Briefly, a 3D-localized fluorescence 'guide star' was formed in the retina via two-photon excitation and scanned over a user-defined 2D area with galvanometer scanning mirrors. The emitted fluorescence was collected and, after being descanned by the same pair of scanning mirrors, directed to the SH sensor. The now stationary fluorescence wavefront was segmented by a lenslet array and focused onto a camera, forming an SH image composed of an array of foci (*Figure 1A*, inset 1). Local phase slopes of wavefront segments were calculated from the displacements of the foci from those taken without aberrations. Assuming spatially continuous aberrations, we computationally reconstructed the wavefront from the phase slopes (*Panagopoulou and Neal, 2005*). We then applied a corrective wavefront, opposite to the measured aberrations, to the DM by controlling the tip, tilt, and piston of each segment (*Figure 1A*, inset 2; *Figure 1—figure supplement 3*) so that mouse ocular aberrations could be canceled out, ensuring diffraction-limited focusing of the two-photon excitation light on the mouse retina.

All in vivo imaging experiments were conducted in anesthetized mice with dilated pupil (Materials and methods). In most experiments, an area of $19 \times 19\ \mu m^2$ of the retina was scanned for 3–10 s for wavefront sensing. To estimate the spatial resolution of our AO-2PFM for in vivo mouse retinal imaging, we imaged Thy1-GFP line M transgenic mice that had green fluorescent protein (GFP) expressed in a subset of RGCs (*Feng et al., 2000*). The image taken without AO showed dim and distorted RGC axons; after aberration correction, we achieved an 8.6× increase in signal and proper visualization of the fine RGC axons (*Figure 1B*). The spatial frequency space representations of the images indicated that AO enhanced the ability of the imaging system to acquire higher resolution information and led to a lateral resolution that was better than ~0.8 μm (*Figure 1B*, insets). For some thin RGC processes (*Figure 1C*), restoring diffraction-limited resolution led to an increase in signal (by 10.6×) and contrast (*Figure 1C, i*), and, for the first time, enabled in vivo 2PFM visualization of varicosities resembling synaptic structures in the mouse retina (*Figure 1C*, white arrowheads). From the axial profile of a thin process (*Figure 1C, ii*), we estimated the axial resolution after AO correction to be 6.7 μm. Both the lateral and axial resolution estimations were close to the theoretical diffraction-limited resolution for a fully-dilated mouse eye with 0.49 numerical aperture (*Geng et al., 2011*).

## AO improves in vivo imaging of retinal vasculature

Retinal vasculature supports the physiological functions of the retina. Retinal vascular diseases can lead to vision loss. Abnormalities in retinal vasculature morphology and physiology serve as important biomarkers for various cerebral and systemic diseases (*Patton et al., 2005*; *Frost et al., 2013*; *Ikram et al., 2013*; *Liew et al., 2008*). Therefore, in vivo characterization of retinal vasculature, especially at the microvasculature level, is of great physiological and clinical importance. Utilizing either confocal microscopy (*Biss et al., 2007*; *Wahl et al., 2019*) or 2PFM (*Qin et al., 2020*; *Bar-Noam et al., 2016*; *Wang et al., 2021*), previous publications have achieved in vivo visualization of retinal microvasculature through either full correction of the mouse eye aberrations (*Biss et al., 2007*; *Wahl et al., 2019*; *Qin et al., 2020*), partial correction of the anterior optics of the mouse eye (*Wang et al., 2021*), or stringent selection of imaging lenses (*Bar-Noam et al., 2016*). These prior demonstration-of-principle experiments suggest that in order to image retinal microvasculature in vivo, mouse eye aberrations need to be corrected, either fully or partially. With our optimized imaging system, we aimed to determine whether aberration correction was indeed essential for visualizing microvasculature. Furthermore, we proceeded to systematically characterize the spatial dependence of mouse eye aberrations and how large a FOV can benefit from a single AO correction.

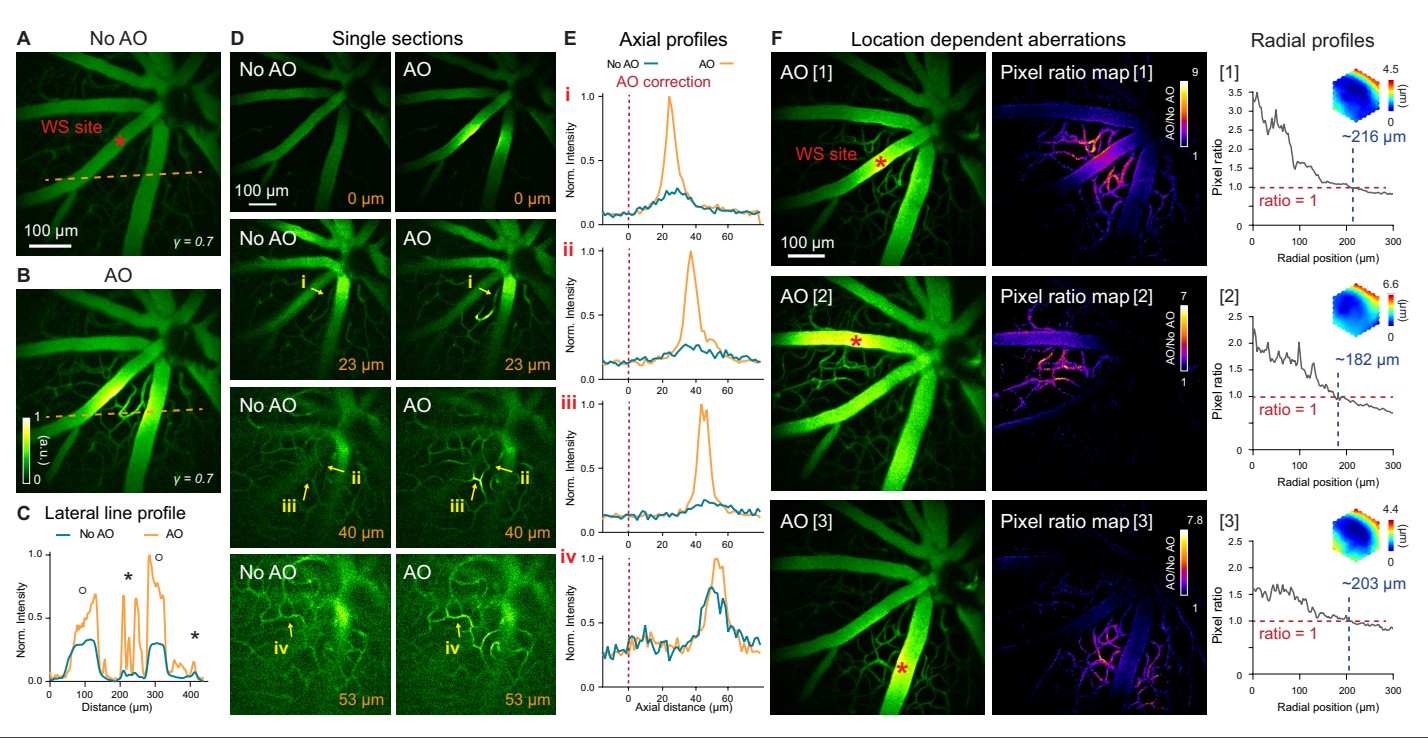

**Figure 2.** In vivo imaging of mouse retinal vasculature with AO-2PFM. (**A,B**) MIPs of image stacks (580×580×128 µm³) of vasculature measured (**A**) without and (**B**) with AO, respectively, normalized to AO image. Red asterisk: center of 19×19 µm² wavefront sensing (WS) area. Gamma correction: 0.7. Representative data from >25 experiments (technical replicates). (**C**) Lateral line profiles along orange dashed lines in A and B. Black circles: large vessels; black asterisks: capillaries. (**D**) Single image planes at 0, 23, 40, and 53 µm below the superficial vascular plexus acquired without and with AO correction performed at the superficial plexus (0 µm), normalized to AO images. (**E**) Axial profiles of capillary structures (**i-iv** in D). Red dashed lines: depth of wavefront sensing area. (**F**) Left: MIPs of image stacks (580×580×110 µm³) acquired with WS performed at different locations in the FOV (red asterisks). Middle: AO/No AO pixel ratio maps. Right: radially averaged profiles of pixel ratio maps, centered at WS sites. Insets: corrective wavefronts. MIPs and pixel ratio maps individually normalized.

The online version of this article includes the following video and source data for figure 2:

**Source data 1.** Source image stacks of retinal vasculature (*Figure 2A, B and D*).

**Source data 2.** Source image stacks with AO measured at different locations (*Figure 2F*).

**Figure 2—video 1.** In vivo 2-photon image stacks of retinal vasculature in a wildtype mouse measured without and with AO performed at different locations in the field of view (red asterisks).

https://elifesciences.org/articles/84853/figures#fig2video1

To verify the necessity of AO in resolving mouse retinal microvasculature and characterize mouse eye induced aberrations, we performed in vivo 2PFM angiography by retro-orbitally injecting dextran-conjugated fluorescein isothiocyanate (FITC) into the non-imaged eye. Aberrations were measured with fluorescence emitted from vessels in the superficial plexus (red asterisk, *Figure 2A*; wavefront sensing area: 19×19 µm²). After AO correction, we observed a 2–10× enhancement in signal (*Figure 2B and C*). Comparing the line signal profiles (along the orange dashed lines, *Figure 2A and B*), we found that AO improved signal for all vessels while its impact on signal of smaller capillaries (*Figure 2C*, black asterisks; 6–10× improvement) was more substantial than on larger vessels (*Figure 2C*, black circles; 2–3× improvement). Despite the substantial signal improvements enabled by AO, we found that most capillaries, due to their size and sparse distribution in space, could be resolved in 3D without AO by our optimized 2PFM, albeit at reduced contrast and resolution (*Figure 2D and E*). Our results indicate that a properly designed 2PFM is capable of acquiring retinal angiograms at the level of individual capillaries.

We further evaluated how the mouse ocular aberrations varied with imaging depth and field position. We found that AO performed at the superficial plexus was beneficial for imaging deeper layers, with the correction at superficial depth improving signal, resolution, and contrast of deeper

vasculature (*Figure 2D and E*). This result indicated that most aberrations of the mouse eye arose from cornea and crystalline lens, instead of retina. Because the crystalline lens of the mouse eye has a gradient refractive index distribution (*Campbell and Hughes, 1981*; *Remtulla and Hallett, 1985*), ocular aberrations should also be field dependent (*Wang and Ji, 2012*; *Wang and Ji, 2013*). Field-dependent aberrations might also be introduced when the mouse eye was positioned off-axis with respect to the eye imaging module. We therefore examined how aberrations varied with FOV position and characterized the area within which a single correction led to substantial signal improvement. We performed AO at different locations of the superficial plexus in the FOV (*Figure 2F*, left column, red asterisks; *Figure 2—video 1*) and compared their performance. The 'AO/No AO' pixel ratio maps (*Figure 2F*, middle column) exhibited field-dependent signal increase with larger gain achieved at pixels closer to the locations of aberration measurements. We quantified the effective area of AO in terms of signal improvement by calculating the radially averaged profiles of these pixel ratio maps (*Figure 2F*, right column; origins at the wavefront sensing locations). We found signal improvement ('AO/No AO' pixel ratio ≥1) within a radius of ~216 μm when AO was performed at the FOV center of this mouse (*Figure 2F*, [1]). For off-center locations, this radius was slightly smaller (*Figure 2F*, [2] and [3]).

## AO enables 3D cellular resolution imaging of neurons in the mouse retina

The mouse retina consists of multiple layers of neurons with different cell types and distinct physiological properties. In the early stage of retinal diseases, abnormal morphology and function are usually confined to specific cell types within a single layer (*Hoon et al., 2014*). Therefore, for microscopic investigations of retinal physiology and pathology, it is essential to resolve cells in 3D. We evaluated whether our optimized 2PFM was capable of 3D cellular resolution imaging without correcting the severe aberrations of the mouse eye.

For this purpose, we imaged the densely fluorescent Thy1-YFP-16 mouse retina in vivo, where all bipolar cells, amacrine cells, and retinal ganglion cells were labeled with yellow fluorescence protein (*Feng et al., 2000*) (YFP). A single AO correction acquired by scanning a 19×19 μm$^2$ area centered on the red asterisk in (*Figure 3A*) substantially improved signal and resolution (*Figure 3A and B*; *Figure 3—video 1*). 2D Fourier transforms of these retinal images indicated that AO recovered higher spatial frequency information (i.e. farther away from the center of *Figure 3C and D*) thus improved both lateral and axial resolution. The resolution enhancement was especially striking along the axial direction, allowing retinal layers to be more clearly differentiated by better resolving neurons at different depths (*Figure 3A and B*, XZ images). This improvement in axial resolution is especially important for functional imaging, because it minimizes neuropil contamination and ensures accurate characterization of the functional properties of neurons (*Ji et al., 2012*; *Wang et al., 2014a*; *Sun et al., 2016*). Therefore, AO was necessary for 3D cellular resolution imaging of retinal neurons in vivo. In the lateral image planes, our optimized 2PFM design and mouse preparation allowed the identification of individual neurons without AO, albeit at lower signal and poorer resolution than those achieved with AO, for inner nuclear layer, inner plexiform layer, and ganglion cell layer (*Figure 3E*). In contrast, subcellular processes could not be clearly visualized without aberration correction (e.g. processes in the inner plexiform layer, *Figure 3E*, white boxes in the middle column; more examples in *Figure 3—figure supplement 1*).

Similar to our vascular imaging results, the Thy1-YFP-16 mouse eye exhibited field-dependent aberrations. For areas away from the AO measurement location (e.g. blue dashed box in *Figure 3A*), although resolution improvement remained, the correction acquired at the FOV center (*Figure 3F*, Central AO) did not increase signal strength as much as the locally acquired correction (centered on the blue asterisk in *Figure 3A*; *Figure 3F*, Local AO). For the Thy1-YFP-16 mouse, the effective area of AO performed at the FOV center was estimated from the 'AO/No AO' ratio map (*Figure 3G*) to have a radius of ~185 μm (*Figure 3H*).

## Strategy for enlarging the effective area of AO correction for 3D cellular resolution imaging

Imaging retinal vascular and neuronal structures, we found that the spatially varying aberrations of the mouse eye limited the effective area for AO correction that was acquired by sensing wavefront from

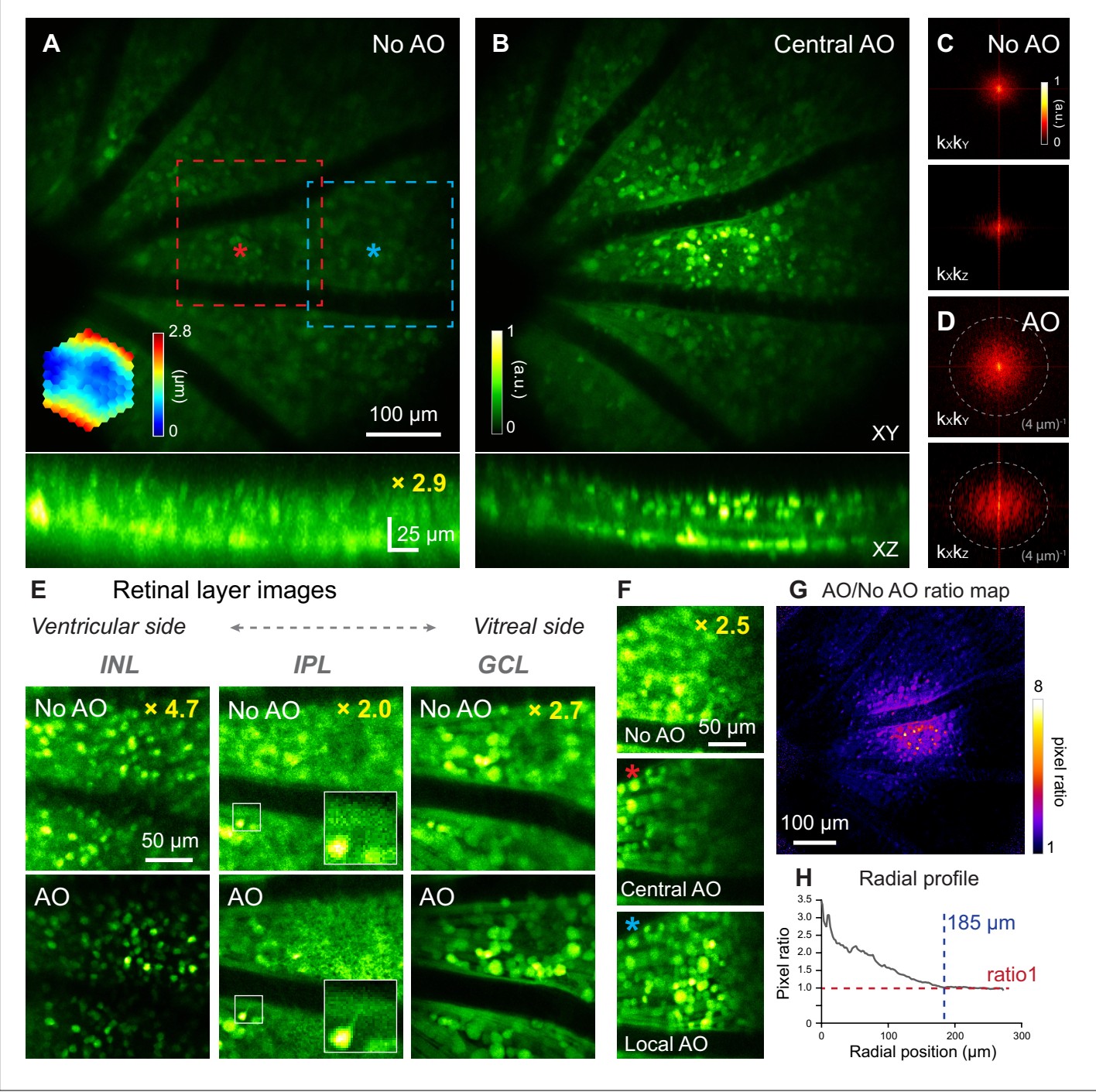

**Figure 3.** In vivo imaging of mouse retinal neurons with AO-2PFM. (**A,B**) MIPs of image stacks (580×580×80 μm³) of a Thy1-YFP-16 retina, measured (**A**) without and (**B**) with AO, respectively, normalized to AO images. Red asterisk: center of a 19×19 μm² WS area. Top: lateral (XY) MIPs. Bottom: axial (XZ) MIPs; 'No AO' image brightness artificially increased by 2.9× for visualization. Representative data from >10 experiments (technical replicates). (**C,D**) $k_X k_Y$ and $k_X k_Z$ spatial frequency space representation of images in (**A,B**). (**E**) Images of different retinal layers within the red dashed box in A acquired (top) without and (bottom) with AO, respectively, normalized to AO images. INL: inner nuclear layer; IPL: inner plexiform layer; GCL: ganglion cell layer. INL/GCL: MIPs of 4.9/7.8-μm-thick image stacks; IPL: single image plane. 'No AO' image brightness artificially increased for visualization (gains shown in each image). White boxes: zoomed-in views. (**F**) Single image planes in GCL at FOV edge (blue dashed box in A) acquired (top) without AO, (middle) with central AO (WS area centered at red asterisk in A), and (bottom) with local AO (WS area centered at blue asterisk in A), respectively. Images normalized to local AO image. 'No AO' image brightness artificially increased by 2.5× for visualization. (**G**) AO/No AO pixel ratio map. (**H**) Radially averaged profile of pixel ratio map, centered at red asterisk in A.

*Figure 3 continued on next page*

*Figure 3 continued*

The online version of this article includes the following video, source data, and figure supplement(s) for figure 3:

**Source data 1.** Source image stacks of retinal neurons (*Figure 3A and B*).

**Source data 2.** Source image stacks of retinal neurons (*Figure 3E*).

**Source data 3.** Source image stacks of retinal neurons (*Figure 3F*).

**Figure supplement 1.** AO improves the visualization of subcellular features in vivo.

**Figure 3—video 1.** In vivo 2-photon image stacks of retinal neurons in a Thy1-YFP-16 mouse measured without and with AO.

https://elifesciences.org/articles/84853/figures#fig3video1

a small region of the retina (e.g. 19×19 μm$^2$ for *Figures 1–3*). Although this approach succeeded in resolving varicosities (*Figure 1C*) and neuronal processes (*Figure 3E and F*), for applications requiring 3D neuronal population imaging, synaptic resolution can be sacrificed in favor of cellular resolution imaging capability over larger FOVs. The latter can be achieved by correcting only for global mouse eye aberrations measured by scanning a larger retinal region for wavefront sensing.

As a demonstration, for a 580×580 μm$^2$ FOV, we measured aberrations from areas of 19×19, 95×95, 190×190, and 380×380 μm$^2$ (*Figure 4A, i-iv*, yellow dashed boxes) and obtained differing corrective wavefronts resulting from the spatially varying aberrations. Quantifying and comparing AO effectiveness by their 'AO/No AO' pixel ratio maps (*Figure 4A*), we found that correcting aberrations from smaller areas provided greater local signal improvement but exhibited faster decay in signal improvement over distance (*Figure 4B, i and ii*). This was because the corrective wavefront acquired from a small FOV completely cancelled out the local aberrations and led to diffraction-limited imaging of local structures. For structures away from the wavefront sensing region and thus experiencing different aberrations, however, the same corrective wavefront led to substantial residual aberrations that degraded AO performance. In contrast, correcting aberrations from a larger area reduced signal improvement in the center of the area but enlarged the overall area within which signal was enhanced, which now extended over the entire imaging FOV (*Figure 4B, iii and iv*). Here, the wavefront measured from scanning the guide star over a larger FOV averaged out the local variations and represented the wavefront distortions common to all field positions. As a result, even though the improvement at the center of the wavefront sensing area was not as large, by removing the common aberrations from the entire FOV, this approach led to a larger effective area for AO correction.

Importantly, this approach enabled large-scale imaging of the retina with 3D cellular resolution, as indicated by retinal cell images taken from the center and edge locations (*Figure 4C*). A more localized wavefront correction (e.g. AO [i], *Figure 4C*) gave rise to brighter and sharper images at the scanning center (*Figure 4C*, insets a; *Figure 4D*, line profiles for a), while a more global wavefront measurement (e.g. AO [iv], *Figure 4C*) benefited more the visualization of neurons towards the edge of the FOV (*Figure 4C*, insets b and c; *Figure 4D*, line profiles for b). Moreover, with global corrections, neuronal images at the center of the area maintained cellular resolution despite reduction in signal gain (*Figure 4C*, insets a; *Figure 4D*, orange line profiles for a). Our results suggest that for diffraction-limited imaging of fine structures within a small FOV, a localized wavefront measurement is required, whereas a global wavefront measurement is preferable for 3D cellular resolution imaging over large FOVs.

## High-resolution in vivo identification of abnormal capillaries in a pathological mouse model

Having demonstrated the effectiveness of our AO-2PFM in improving signal, contrast, and spatial resolution for in vivo retinal imaging, we utilized our system to study retinal microvascular pathology. Retinal angiomatous proliferation (RAP), a subtype of age-related macular degeneration, is characterized by capillary proliferation that originates from the sensory retina and extends into the subretinal space (*Yannuzzi et al., 2001*). Replicating the characteristic phenotypes of human RAP, a transgenic mouse model, the very low-density lipoprotein receptor knockout (VLDLR-KO) mouse, has been employed to study the underlying mechanism of RAP. In this model, the gene encoding VLDLR, which mediates anti-angiogenic signaling in retinal vasculature, is knocked out, leading to overgrown intra-retinal vasculature and subretinal neovascularization (*Heckenlively et al., 2003*; *Hu et al., 2008*). In addition, fluorescein angiography revealed that the VLDLR-KO model of proliferative vascular

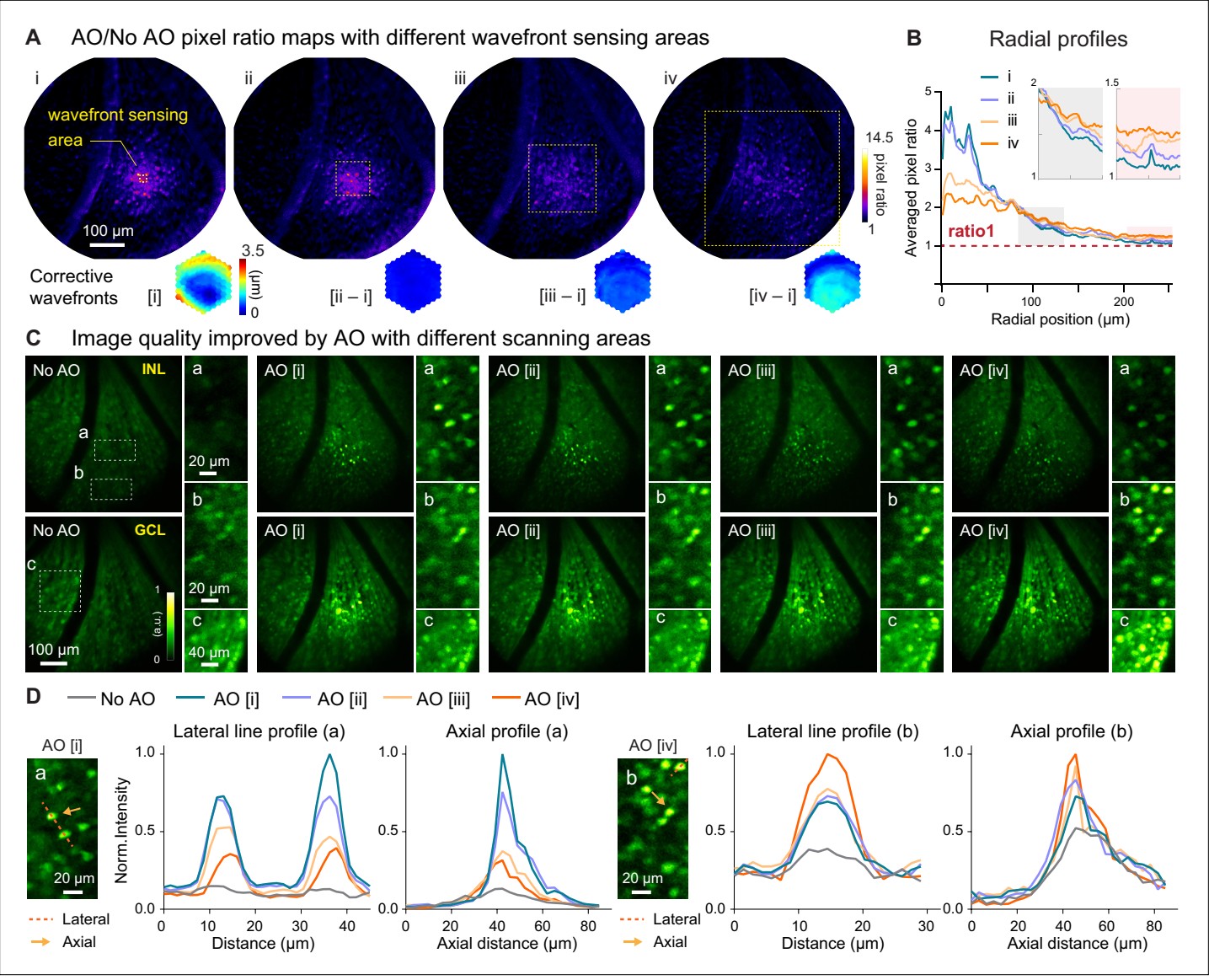

**Figure 4.** Larger WS areas enlarges the effective region of AO correction for 3D cellular resolution imaging. (**A**) Top: AO/NoAO pixel ratio maps for corrections with differently sized WS areas (yellow dashed boxes; **i**, 19×19 µm²; **ii**, 95×95 µm²; **iii**, 190×190 µm²; **iv**, 380×380 µm²). Bottom: (for [**i**]) corrective wavefront and (for [**ii-iv**]) difference in wavefronts between [**ii-iv**] corrective wavefronts and [**i**] corrective wavefront. (**B**) Radially averaged profiles of pixel ratio maps in (**A**). Insets: zoomed-in views of shaded areas. (**C**) Single image planes acquired (top) from INL and (bottom) GCL without and with AO using corrective wavefronts [**i-iv**], respectively. Insets: zoomed-in views of areas at FOV (**a**) center and (**b,c**) edge. All images normalized to AO images (AO [**i**] for inset a; AO [**iv**] for inset b,c). (**D**) Lateral (along red dashed lines) and axial (at the center of the neurons indicated by orange arrows) profiles of neurons in the (**a**) central and (**b**) edge regions.

The online version of this article includes the following source data for figure 4:

**Source data 1.** Source image stacks with AO by measuring aberrations with different wavefront sensing areas.

retinopathy has extensive focal vascular leakage (*Heckenlively et al., 2003*; *Hu et al., 2008*; *Li, 2007*; *Xia et al., 2013*). However, the lack of sufficient spatial resolution and optical sectioning capability makes it challenging for fluorescence angiography to identify the 3D location and characterize the structure of the vascular lesions in vivo.

We utilized AO-2PFM to image in vivo the retina of VLDLR-KO/Sca1-GFP and their wildtype control WT/Sca1-GFP mice, both with vascular endothelial cells in the retina labeled with GFP (*Xia et al., 2013*). In order to detect microscopic capillary pathology, we used 19×19 µm² wavefront sensing area to achieve diffraction-limited imaging performance, which led to high-resolution images of endothelial

cell linings of retinal vessels in both mouse lines (*Figure 5A and B*). Interestingly, in the VLDLR-KO/Sca1-GFP retina, images acquired with AO revealed a disruption in the capillary endothelium labeled by GFP, where the endothelial cells lined the walls of a short capillary branch but not its end face, leading to a ring-like structure (*Figure 5A and C*, yellow asterisks, insets i-ii; *Figure 5—video 1*). Such ring-like structures were not observed in the WT/Sca1-GFP retina (*Figure 5B*). We further confirmed the presence of such microvascular lesions using ex vivo 2PFM imaging of dissected VLDLR-KO/Sca1-GFP retinas (*Figure 5—figure supplement 1A and C*; *Figure 5—video 2*). Whereas similarly structured capillary disruptions were observed in the VLDLR-KO/Sca1-GFP retina, consistent with the in vivo investigation, capillaries in the wildtype control had endothelial cells fully enclosing both the walls and the ends of capillary branches (*Figure 5—figure supplement 1B and D*; *Figure 5—video 3*).

We hypothesized that these lesions as capillary disruptions observed in the VLDLR-KO/Sca1-GFP retina were the locations of dye leakage. To test this hypothesis, enabled by AO, we first located a microvascular lesion in a VLDLR-KO/Sca1-GFP mouse retina (*Figure 5C*, orange box, inset). Then we retro-orbitally injected the green fluorescent dye FITC into the non-imaged eye, which labeled the blood plasma within the retinal vasculature (*Figure 5D*). Immediately after dye injection, we observed dye leakage around the lesion site (*Figure 5D*, orange dashed area). A control experiment was carried out by introducing FITC into the healthy WT/Sca1-GFP mouse retina retro-orbitally, where neither capillary disruptions nor dye leakage were observed (*Figure 5E*).

To further study the association between dye leakage and microvascular lesions, we injected the NIR dye Evans Blue (EB) into the retinal vasculature and performed dual-color two-photon imaging of the VLDLR-KO retina. Similar to the experiments with FITC, we observed leakage in the knockout mouse retina, with EB persistently staining retinal tissue and the stained volume expanding over 3 days of consecutive imaging (*Figure 5F*). We observed capillary lesions (*Figure 5F*, insets) in the stained volume, suggesting a spatial correlation between dye leakage and capillary abnormalities. Moreover, on the third day, we observed GFP-positive cells within the dye-stained retinal volume that were absent in previous 2 days (*Figure 5G*). The morphology of these cells resembled that of activated microglia (*Joseph et al., 2021*; *Ozaki et al., 2022*). We speculated that the leakage of EB triggered local immune response and recruited ocular immune cells to the impacted area. With the subcellular resolution provided by AO-2PFM, we were able to track dynamic changes in the processes of the same cell over time (*Figure 5G*, bottom). Control experiment in WT/Sca1-GFP retina showed local small-scale EB leakage (*Figure 5—figure supplement 2*), probably resulting from normal remodeling of the retinal vasculature (*Selvam et al., 2018*). Our findings revealed, for the first time, the microscopic morphological details of vasculature lesions and suggested that these capillary disruptions served as intraretinal origins of vascular leakage in the VLDLR knockout mouse. Here, AO was essential for 2PFM to achieve high-resolution identification and characterization of microvasculature lesions in vivo. Together with our optimized sample preparation, AO-2PFM also allowed us to track these lesions, dye leakage, and associated immune response longitudinally, making it possible to investigate the development and progression of vasculature-associated diseases at subcellular resolution in vivo.

## High-resolution in vivo imaging of retinal pharmacology

With the 3D cellular resolution imaging capability enabled by AO-2PFM, we can now image the functional activity of retinal neurons with high fidelity in healthy or diseased retina in vivo using activity sensors such as the genetically encoded calcium indicator GCaMP6s (*Chen et al., 2013*).

As a demonstration, we studied how pharmacological manipulation affects RGC activity in vivo in a mouse model of retinal degeneration. As the afferent neurons of the retina, RGCs deliver retinal circuit output to the rest of the brain and play a crucial role in visual perception. RGCs in the *rd1* mouse, the oldest and most widely studied animal model of retinal degeneration (*Chang et al., 2002*), become hyperactive after photoreceptor death caused by a mutation in the *Pde6b* gene (*Sekirnjak et al., 2011*; *Telias et al., 2019*). Recent studies have suggested that RGC hyperactivity masks light-evoked signals initiated by surviving photoreceptors and impedes remaining light-elicited behaviors (*Telias et al., 2019*; *Telias et al., 2022*). Studying RGC hyperactivity therefore is of great importance both for understanding the pathology of retinal degeneration and for developing pharmacological therapies (*Cao et al., 2021*). However, RGC hyperactivity has been only studied ex vivo on dissected retinas

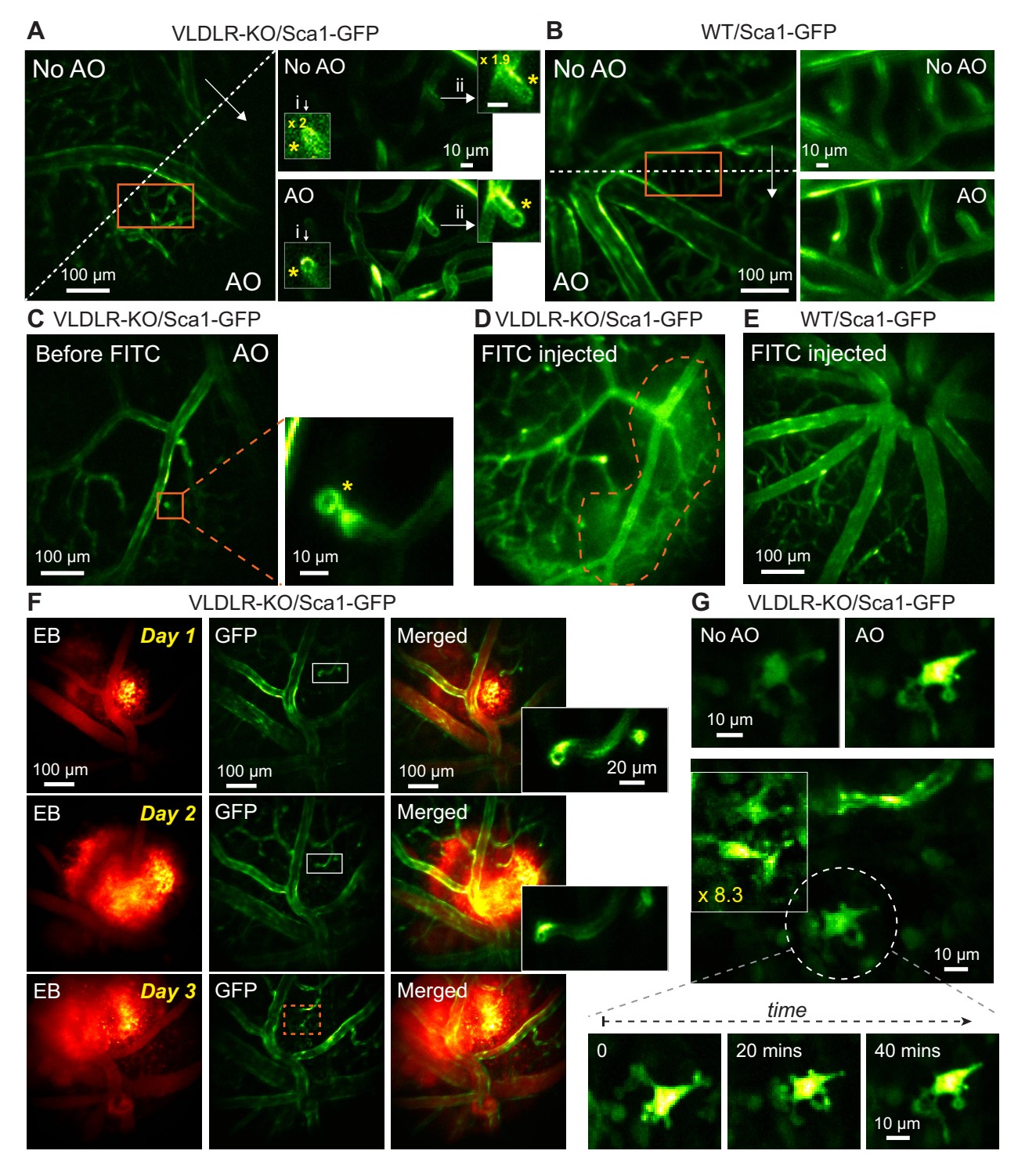

**Figure 5.** In vivo vasculature imaging in pathological and healthy retinas. (**A,B**) Left: MIPs of image stacks of (**A**) VLDLR-KO/Sca1-GFP ($580\times580\times94\ \mu m^3$) and (**B**) WT/Sca1-GFP ($520\times520\times120\ \mu m^3$) mouse retinas, measured (arrow start) without and (arrow end) with AO. Asterisks: capillary disruptions. Insets: zoomed-in views individually normalized for better visualization. 'No AO' inset brightness artificially increased for visualization (gains shown in inset). (**C**) A single image plane of a VLDLR-KO/Sca1-GFP mouse retina before FITC injection. Inset: MIP of a zoomed-in image stack ($58\times58\times8.2\ \mu m^3$) showing capillary lesion (orange box). (**D**) The same FOV in (**C**) after FITC injection. Dashed region: area with heightened fluorescence outside the vasculature. (**E**) MIP of an image stack ($580\times580\times150\ \mu m^3$) of a WT/Sca1-GFP mouse retina after FITC injection. (**F**) Retinal images taken on (top) day 1, (middle) day 2, and (bottom) day 3 after Evans Blue (EB) injection. Left: near-infrared channel showing EB-labeled vasculature and tissue staining (MIP of a $580\times580\times166$

*Figure 5 continued on next page*

*Figure 5 continued*

μm³ volume). Middle: green channel showing GFP-labeled vasculature (single planes). Right: merged images. Insets: zoomed-in views of white boxes within the GFP images. (**G**) Microglia observed in EB-injected VLDLR-KO/Sca1-GFP mouse retina on day 3 near the lesion site (orange dashed box in **F**). Top: microglia imaged without and with AO. Middle: multiple microglia in the leaking region. Signal in the boxed region was artificially increased by 8.3× for visualization. Bottom: time-lapse images of the microglia in white dashed circle. All images are single planes. Wavefront sensing area: 19×19 μm². In vivo data in this Figure were obtained from 3 VLDLR-KO and 2WT mice (biological replicates).

The online version of this article includes the following video, source data, and figure supplement(s) for figure 5:

**Source data 1.** Source image stacks of VLDLR-KO/Sca1-GFP mouse retina (***Figure 5A***, full FOV).

**Source data 2.** Source image stacks of WT/Sca1-GFP mouse retina (***Figure 5B***, full FOV).

**Source data 3.** Source image stacks of VLDLR-KO/Sca1-GFP mouse retina (***Figure 5C***, full FOV).

**Source data 4.** Source image stack of VLDLR-KO/Sca1-GFP mouse retina after FITC injection (***Figure 5D***).

**Source data 5.** Source image stack of WT/Sca1-GFP mouse retina after FITC injection (***Figure 5E***).

**Source data 6.** Source image stacks of VLDLR-KO/Sca1-GFP mouse retina after EB injection (***Figure 5F***).

**Figure supplement 1.** Ex vivo 2PFM imaging of dissected VLDLR-KO/Sca1-GFP and WT/Sca1-GFP mouse retinas.

**Figure supplement 2.** In vivo AO-2PFM imaging of Evans Blue (EB) leakage in healthy retina.

**Figure 5—video 1.** In vivo two-photon image stacks of abnormal retinal capillaries in a VLDLR-KO/Sca1-GFP mouse measured without and with AO.
https://elifesciences.org/articles/84853/figures#fig5video1

**Figure 5—video 2.** Ex vivo two-photon image stacks of abnormal capillaries in dissected VLDLR-KO/Sca1-GFP mouse retinas.
https://elifesciences.org/articles/84853/figures#fig5video2

**Figure 5—video 3.** Ex vivo two-photon image stacks of normal capillaries in dissected WT/Sca1-GFP mouse retinas.
https://elifesciences.org/articles/84853/figures#fig5video3

(***Telias et al., 2019***; ***Cao et al., 2021***), preventing longitudinal evaluation of degeneration progression and therapeutic approaches.

Here, we characterized RGC hyperactivity in vivo and studied the effect of Lidocaine, a use-dependent Na⁺ channel blocker, on alleviating hyperactivity of RGCs in the *rd1*-Thy1-GCaMP6s mouse using AO-2PFM and calcium imaging. The *rd1*-Thy1-GCaMP6s mice selectively express GCaMP6s in their RGC layer of the retina (***O'Brien et al., 2014***; ***Chen et al., 2012***). Because RGC hyperactivity is usually studied by ex vivo tools such as multi-electrode array (MEA) or single cell electrophysiology recordings, to establish the calcium signature of RGC hyperactivity, we first carried out simultaneous cell-attached and ex vivo 2PFM calcium recordings of the same hyperactive alpha RGCs in a dissected *rd1* mouse retina (***Figure 6A***). Consistent with previous reports on ex vivo retina (***Telias et al., 2019***; ***Telias et al., 2022***; ***Cao et al., 2021***), RGC hyperactivity was observed as high-frequency action potentials. In terms of calcium signaling (quantified as calcium response magnitude ΔF/F, with F being baseline brightness and ΔF being the difference from baseline brightness), hyperactivity measured ex vivo was correlated with a heightened ΔF/F of the GCaMP6s-expressing soma. A temporally varying firing rate led to transient fluctuations in its calcium signal (ROI 1, ***Figure 6A***), whereas a sustained high firing rate led to heightened fluorescence brightness without obvious transients (ROI 2, ***Figure 6A***). After ~20 seconds of 2% Lidocaine bath perfusion, spontaneous spiking from the RGC was largely suppressed with a ΔF/F close to 0. After artificial cerebrospinal fluid (ACSF) washout, RGC hyperactivity partially recovered, which was associated with an increase of ΔF/F magnitude. The observed time course and suppressive effect of Lidocaine application on RGC hyperactivity were consistent with ex vivo multi-electrode array (MEA) recordings (***Figure 6—figure supplement 1A and B***). The characteristics of the corresponding calcium responses were also observed in 2-photon population imaging of multiple RGCs in dissected *rd1* retina (***Figure 6B***), with the brightness and ΔF/F of the GCaMP6s-expressing neurons reduced by Lidocaine application and followed by partial or full recovery after washout.

Having confirmed that RGC hyperactivity was associated with heighted calcium levels, we next performed AO-2PFM calcium imaging to directly study how Lidocaine affected RGC hyperactivity in vivo. Through the *rd1*-Thy1-GCaMP6s mouse eye, AO increased RGC brightness by on average 4× and enabled high-resolution visualization of both RGC somata and their processes (***Figure 6C***). The signal increase enabled by AO was particularly important for the *rd1*-Thy1-GCaMP6s mouse, because the RGCs here had dimmer fluorescence than the other lines that we investigated. For these RGCs,

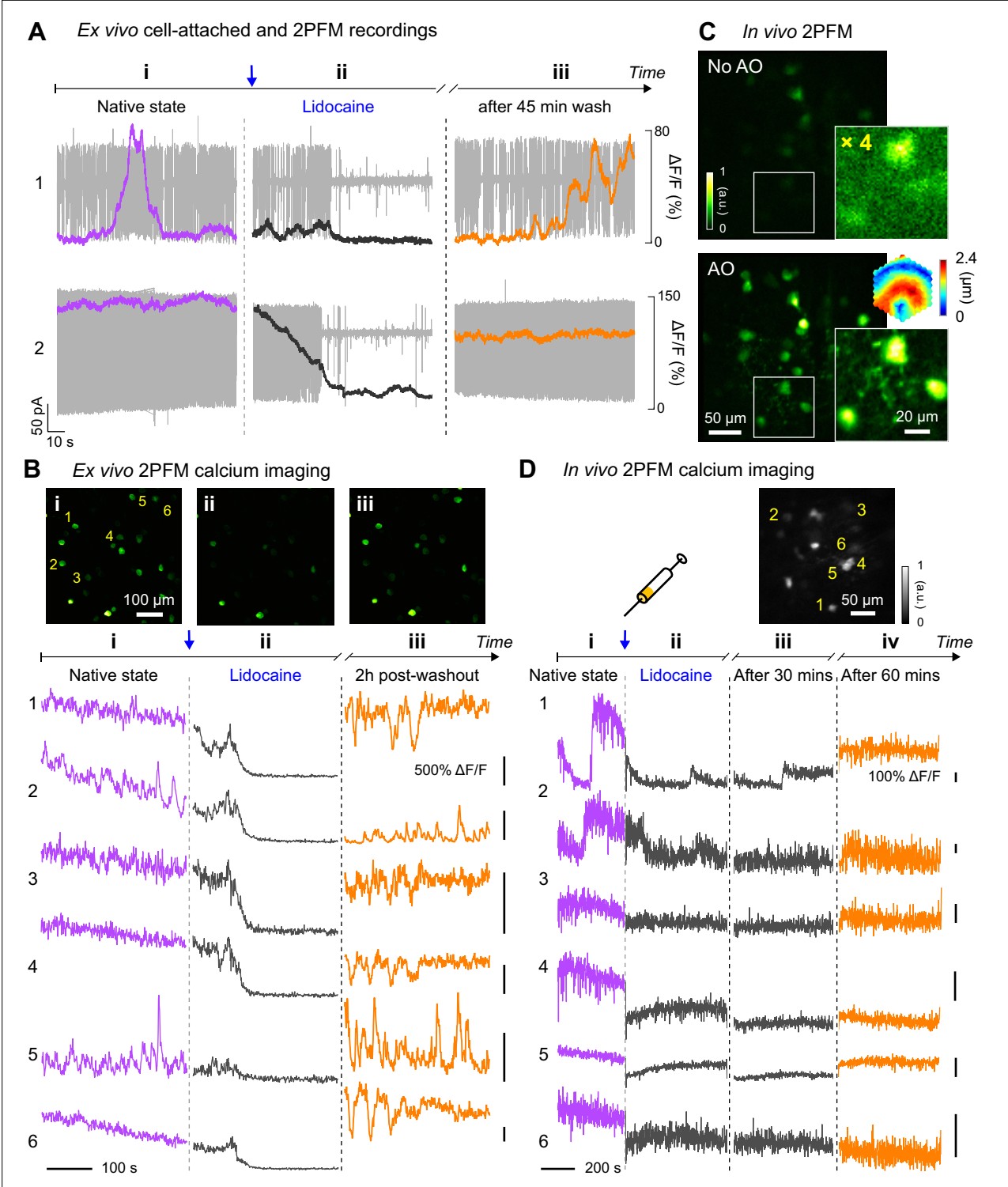

**Figure 6.** In vivo calcium imaging of Lidocaine-suppressed RGC hyperactivity in *rd1*-Thy1-GCaMP6s mouse retina. (**A**) Simultaneous cell-attached and 2PFM calcium recordings of two RGCs before, during, and 45 min after Lidocaine treatment. Representative data from >3 cells. (**B**) Top: average intensity projections of ex vivo 2PFM images of RGCs in a dissected retina (**i**) before, (**ii**) right after, and (**iii**) 2 hr after Lidocaine treatment, normalized to the left image. Bottom: Ex vivo calcium dynamics of 6 RGCs therein. Representative data from >3 retinas. (**C**) In vivo single image planes of RGCs acquired without and with AO, respectively, normalized to AO images. Insets: zoomed-in views and corrective wavefront; 'No AO' inset brightness artificially increased by 4.0× for visualization. Representative data from >3 retinas. (**D**) In vivo calcium dynamics of 6 RGCs (**i**) before, (**ii**) right after, (**iii**) 30 minutes after, and (**iv**) 60 minutes after Lidocaine treatment, respectively. Wavefront sensing area: 19×19 µm².

*Figure 6 continued on next page*

*Figure 6 continued*

The online version of this article includes the following source data and figure supplement(s) for figure 6:

**Source data 1.** Source data of ex vivo cell-attached and 2PFM recordings (*Figure 6A*): RGC #1.

**Source data 2.** Source image sequences of ex vivo 2PFM calcium imaging (*Figure 6B*).

**Source data 3.** Source images of *rd1*-Thy1-GCaMP6s mouse retina (*Figure 6C*, full FOV).

**Source data 4.** Source data for in vivo 2PFM calcium imaging (*Figure 6D*).

**Figure supplement 1.** Ex vivo multielectrode array (MEA) recordings of Lidocaine-modified RGC hyperactivity in *rd1*-Thy1-GCaMP6s mouse retina.

correcting the eye-induced aberration was essential for their visualization and high-fidelity functional investigations at cellular resolution in vivo. To maximize the fluorescence signal, we performed AO with a small (19×19 μm²) wavefront sensing area. Before injecting Lidocaine, we observed slow fluctuations in the brightness of GCaMP6s-expressing RGCs (*Figure 6D, i*), similar to the slow dynamic events in RGC calcium traces measured ex vivo. One minute after retro-orbital injection of Lidocaine into the non-imaged eye, hyperactivities from these cells were substantially inhibited for an hour as indicated by the reduction of RGC GCaMP6s fluorescence brightness (*Figure 6D, ii and iii*). Imaging the same RGCs 60 min after injection (*Figure 6D, iv*), we detected partial recovery of RGC brightness, consistent with our ex vivo recordings after washing out. Here, by studying the suppression effects of Lidocaine on RGC hyperactivity within living mice, we demonstrated that AO-2PFM can monitor the pathology and pharmacology of retinal diseases at high resolution in vivo.

## Discussion

By optimizing optical design and sample preparation and using direct wavefront sensing AO to correct mouse eye aberrations, we demonstrated here the first in vivo visualization of putative retinal synaptic structures, the first in vivo identification of capillary lesions with sub-capillary details, and the first in vivo detection of RGC hyperactivity and its suppression by pharmacological reagents.

To image mouse retina in vivo with 2PFM, one can either utilize a standard objective lens (*Palczewska et al., 2014*; *Bar-Noam et al., 2016*) or the mouse eye's optics itself (*Sharma et al., 2013*; *Qin et al., 2020*) to focus the excitation light and collect the fluorescence emission. The former approach requires long-working-distance objective lenses and, more importantly, suffers from severe aberrations caused by the refractive power of the ocular optics (mostly crystalline lens, as cornea was typically flattened in these systems). For this reason, to achieve the best image quality and the largest FOV size, using mouse eye itself as the focusing element as implemented here is preferable.

Among the studies that used the mouse eye optics for imaging, discrepancies exist in how large the mouse ocular aberrations are and how essential AO is for vasculature and cellular imaging in the mouse retina. For the multiple mouse strains investigated here (i.e. wild-type (C57BL/6), Thy1-GFP line M, Thy1-YFP-16, VLDLR-KO/Sca1-GFP, WT/Sca1-GFP, and *rd1*-Thy1-GCaMP6s), we found that their ocular aberrations were typically within the range of 3~5 μm peak-to-valley (P-V; after removing tip, tilt, and defocus) and 0.4~0.8 μm rms (*Figure 1—figure supplement 3*) without notable differences in severity across strains. While consistent with most previously reported values (*Alt et al., 2010*; *Wahl et al., 2019*), our study differs significantly from a recent AO-2PFM study that reported extremely large aberrations (e.g. 12–25 μm P-V, *Qin et al., 2020*) and found it difficult to resolve microvasculature and cell bodies in 2D in vivo without AO. In contrast, with our carefully designed microscope and system aberration correction procedure, we achieved capillary visualization, 2D single-cell resolution, and retinal layer differentiation by only correcting system aberrations ('No AO' in our case). Given that our system aberrations were much smaller than mouse ocular aberrations (*Figure 1—figure supplement 3*), our study indicated that a well-engineered 2PFM like ours should be sufficient for in vivo retinal imaging applications requiring only capillary and 2D cellular resolution.

One factor that strongly impacted image and wavefront sensing quality was the sample preparation procedure. We found that our specially designed 0-diopter contact lens encircled with a supportive flat base were essential for high-quality imaging by 2PFM both without and with AO (*Figure 1—figure supplement 1*). Similar improvement in imaging quality by 0-diopter contact lens was reported previously for in vivo optical coherence tomography imaging of the rat retina, where it was hypothesized that the application of the contact lens smoothed corneal defects and reduced wavefront error of the

anterior segment of the eye (*Liu et al., 2013*). The supportive flat base encircling the optical zone of our contact lens (*Ikeda et al., 2018*) and the gel completely separated the eye surface from air and prevented cataract formation. Together, they enabled high-quality SH images and accurate corrective wavefronts to be acquired throughout the experiment.

Incorporating direct-wavefront-sensing-based AO with 2PFM, we found that location-dependent aberrations led to local improvement in the mouse retina in vivo. To enlarge the high-resolution area enabled by AO, one way is to stitch images from smaller areas, each with its own local AO correction (*Wang and Ji, 2012*; *Wang and Ji, 2013*; *Zhang et al., 2020*). However, this procedure can be time-consuming and thus nonideal for in vivo functional studies. By scanning differently sized areas for wavefront sensing, we identified a trade-off between AO performance (i.e. resolution and signal enhancement) and effective area. We demonstrated that a single corrective wavefront acquired by scanning the guide star over a more extended area led to 3D cellular resolution imaging over a larger retinal volume, simplifying the procedure for future functional studies of neuronal populations in the retina. It is worth noting that, instead of incorporating scan lenses optimized for large scanning angles, our homebuilt 2PFM system utilized regular achromatic doublet lenses, which reduced cost but limited the overall imaging FOV of our microscope. The effective AO area and imaging FOV would be further increased by incorporating high-performance scan lenses.

To study retinal pathologies for physiological and clinical insights, it is ideal to conduct longitudinal investigations in vivo. As demonstrated recently with scanning laser ophthalmoscopy (SLO) (*Li et al., 2020a*), direct visualization of RGCs provides more sensitive readout of RGC degeneration than in vivo retinal characterization techniques such as optical coherence tomography and pattern electroretinogram. SLO has been combined with calcium imaging to characterize the dynamic functional changes of RGCs in disease models (*Yin et al., 2013*; *Cheong et al., 2018*; *Li et al., 2022*). Although the current commercial availability of SLO systems can make them more easily adoptable than our homebuilt AO-2PFM, the visible excitation light utilized in SLO activates photoreceptors, requiring additional considerations in experimental designs. Without correcting the wavefront distortion introduced by the ocular tissue to both excitation and emission light, the resolution of SLO is limited to cellular level. In contrast, the NIR excitation light utilized in 2PFM minimizes photoreceptor activation; aberration correction by AO also enables subcellular resolution to be achieved in vivo.

Importantly, to probe microscopic early-stage pathologies, high spatial resolution is needed. In human applications, investigation of retinal vascular abnormalities are limited to capillary resolution (*Jia et al., 2015*; *Hormel et al., 2021*). Sub-capillary morphology and dynamics of the mouse retina were recently observed by light-sheet microscopy, however with ex vivo preparations (*Prahst et al., 2020*). To our knowledge, sub-capillary features had not been observed in the living mouse eye previously. In this work, applying AO-2PFM, we studied retinal vasculature in a pathological mouse model with proliferative vascular retinopathy at sub-capillary resolution in vivo. Recovering diffraction-limited resolution, AO enabled us to identify capillary lesions as capillary endothelium disruptions that were associated with dye leakage in two-photon fluorescence angiograms. Moreover, the repeatable and reliable AO performance allowed us to track the same retinal region over multiple days and image cell migration at subcellular resolution.

We also applied our AO-2PFM to in vivo activity imaging of RGCs in a mouse model of retinal degeneration. Due to the dimmer brightness of the fluorescence indicator in this model, AO was essential in increasing signal strength and enabling high-sensitivity interrogation of the effects of pharmacological manipulation on RGC hyperactivity. Traditionally, pharmacological effects on retina are studied by electrophysiological and imaging tools on ex vivo retinal preparation, or in vivo by indirect assessments downstream in the visual pathway or through behavior test (*Telias et al., 2022*). Taking retinal degeneration as an example, treatment-induced photosensitization enhancement has been mainly evaluated through electrophysiology or ex vivo imaging of dissected retinas. AO-2PFM enabled us to evaluate how pathological RGC hyperactivity was suppressed by an example pharmacological agent, lidocaine, at single cell level noninvasively. Together with the capability for longitudinal investigations discussed above, we envision that the AO-enabled high-sensitivity subcellular and cellular 2PFM imaging would become a highly enabling technology for pathological and pharmacological investigations of the mouse retina in vivo.

# Materials and methods

## Key resources table

| Reagent type (species) or resource | Designation | Source or reference | Identifiers | Additional information |
|---|---|---|---|---|
| Genetic reagent (*M. musculus*) | C57BL/6 J | Jackson Laboratory | Stock #000664 | |
| Genetic reagent (*M. musculus*) | B6.Cg-Tg(Thy1-YFP)16Jrs/J | Jackson Laboratory | Stock #003709 | |
| Genetic reagent (*M. musculus*) | B6.Cg-Tg(Ly6a-EGFP)G5Dzk/J | Jackson Laboratory | Stock #012643 | |
| Genetic reagent (*M. musculus*) | B6;129S7-Vldlr$^{tm1Her}$/J | Jackson Laboratory | Stock #002529 | |
| Genetic reagent (*M. musculus*) | C3H/HeJ | Jackson Laboratory | Stock #000659 | |
| Genetic reagent (*M. musculus*) | C57BL/6J-Tg(Thy1-GCaMP6s)GP4.3Dkim/J | Jackson Laboratory | Stock #024275 | |
| Software, algorithm | ImageJ software | http://imagej.nih.gov/ij/ | RRID:SCR_003070 | |
| Software, algorithm | GraphPad Prism software | https://graphpad.com | RRID:SCR_015807 | |
| Software, algorithm | MATLAB | https://www.mathworks.com/products/matlab.html | RRID:SCR_001622 | |
| Chemical compound, drug | Lidocaine | Phoenix | NDC: 57319-533-05 | |

## Animal use

All animal experiments were conducted according to the National Institutes of Health guidelines for animal research. Procedures and protocols (AUP-2020-06-13343) were approved by the Institutional Animal Care and Use Committee at the University of California, Berkeley.

## AO two-photon fluorescence microscope (AO-2PFM)

The AO-2PFM was built upon a homebuilt 2PFM (*Figure 1A*) incorporated with a direct-wavefront-sensing-based AO module, as described in detail previously (*Li et al., 2020b*). Briefly, 920 nm output from a femtosecond Ti:Sapphire laser (Coherent, Chameleon Ultra II) was expanded (2×, Thorlabs, GBE02-B) after a Pockel Cell (ConOptics, 350–80-LA-02-BK). The beam was then scanned with a pair of optically conjugated (by L1-L2, FL = 85 mm; Edmund Optics, 49–359-INK) galvanometer mirrors (Cambridge Technology, 6215H). A pair of achromatic lenses (L3-L4, FL = 85 and 300 mm; Edmund Optics, 49–359-INK and 49–368-INK) relayed the galvos to the DM (Iris AO, PTT489). The focal plane position of two-photon excitation in the mouse retina was controlled by an electrically tunable lens (ETL; Optotune, EL-16–40-TC-VIS-5D-C), which was conjugated to the DM (by L5-L6, FL = 175 and 400 mm; Edmund Optics, 49–363-INK and Newport, PAC090). The ETL was then relayed to the pupil of the mouse eye by L7 (FL = 200 mm; Thorlabs, AC254-200-AB) and L8, which was composed of two identical lenses (FL = 50 mm; Thorlabs, AC254-050-AB). The two 50-mm-FL lenses in L8 were used together with a combined FL of 25 mm, and they were mounted with their curved surfaces facing and almost touching each other (*Figure 1—figure supplement 1A and D* ) to minimize aberrations during large-angle scanning. For 2PFM imaging, the emitted fluorescence from the mouse retina was collected by the mouse eye, travelled through L8-L7 and the ETL, reflected by a dichroic mirror (D2; Semrock, Di02-R785–25×36), focused by a lens (L9, FL = 75 mm; Thorlabs, LB1309-A), and detected by a photomultiplier tube (PMT, Hamamatsu, H7422-40). For direct wavefront sensing, D2 was moved out of the light path and the emitted fluorescence was descanned by the galvo pair, reflected by a dichroic mirror (D1; Semrock, Di03-R785-t3-25×36), and relayed to a Shack-Hartmann (SH) sensor by a pair of lenses (L10-L11, FL = 60 and 175 mm; Edmund Optics, 47–638-INK and 47–644-INK). The SH sensor was composed of a lenslet array (Advanced Microoptic System GmbH, APH-Q-P500-R21.1) and a camera (Hamamatsu, Orca Flash 4.0) that was placed at the focal plane of the lenslet array. Wavefront aberrations were measured from the shifts of SH pattern foci, reconstructed with custom MATLAB code, and the corresponding corrective pattern was then applied to the DM.

## System correction

Before imaging the mouse retina, system aberration caused by imperfect and/or misaligned optics was corrected. Due to the path difference between the two-photon illumination and the fluorescence wavefront sensing paths (*Sulai and Dubra, 2014*), system correction was performed with a

modal-based optimization approach (*Wahl et al., 2019*; *Booth, 2014*). Specifically, with 0 mA applied to the ETL, we imaged a fluorescent lens tissue sample at the focal plane of L7 and applied 11 values (–0.1~0.1 µm rms at an increment of 0.02 µm) for each of the first 21 Zernike modes excluding piston, tip, tilt, and defocus. The optimal value for each Zernike mode was determined by maximizing the fluorescence intensity of the sample and it was applied to the DM before proceeding to the next Zernike mode. An SH pattern was obtained with system aberration corrected and was used as the SH reference for calculating sample-induced aberrations. All images taken with system correction were indicated in the main text as 'No AO'.

To change the focal plane within the retina, we varied the electric current applied to the ETL. We characterized how system aberrations varied with the ETL current (*Figure 1—figure supplement 1*). We carried out system correction with 0 mA ETL current applied (*Figure 1—figure supplement 3A*). Additional aberrations introduced by setting ETL current to 20, 40, 60, and 80 mA were negligible (*Figure 1—figure supplement 1B*) compared with eye-induced aberrations (*Figure 1—figure supplement 3*) and minimally affected in vivo imaging (*Figure 1—figure supplement 1C*). We also evaluated how system aberrations varied with the distance D between the mouse eye pupil and the imaging module (*Figure 1—figure supplement 1D*). Using Zemax for ray tracing, we found its effect to be similarly minimal (*Figure 1—figure supplement 1E*). Our typical in vivo retinal imaging was performed with 10~60 mA of ETL currents and 2~4 mm D values (*Figure 1—figure supplement 1D*). Simulating the mouse eye as an ideal lens behind a 0-diopter contact lens (*Figure 1—figure supplement 2A*) made of PMMA (1.49 refractive index) and 0.5-mm-thick eye gel (1.33 refractive index), we calculated the focal shifts and FOVs for different ETL currents using Zemax and found a linear focal shift with ETL current and relatively constant FOV during 3D imaging (*Figure 1—figure supplement 1F*). Imaging FOV and axial shift were determined from Zemax simulation for D=2 mm.

## In vivo imaging

All mice (Wild-type C57BL/6 J and Thy1-YFP-16, the Jackson laboratory; VLDLR-KO/Sca1-GFP and WT/Sca1-GFP, Gong lab; GCaMP6s-*rd1*, Kramer lab) were at least 8 weeks old at the time of imaging. The *rd1*-Thy1-GCaMP6s mice were generated by crossing *rd1* (C3H/HeJ, the Jackson laboratory) with Thy1-GCaMP6s (C57BL/6J-Tg(Thy1-GCaMP6s)GP4.3Dkim/J, the Jackson laboratory). In vivo imaging was carried out on mice under isoflurane anesthesia (~1.0% by volume in $O_2$). Prior to imaging, the mouse pupil was dilated with one drop of 2.5% phenylephrine hydrochloride (Paragon BioTeck, Inc) and one drop of 1% tropicamide (Akorn, Inc). A 0-diopter customized rigid contact lens (*Figure 1—figure supplement 2A*, Advanced Vision Technologies) was placed on the eye, with eye gel (Genteal) applied in between the eye and the contact lens to prevent cornea drying and clouding. Excessive eye gel was removed by gently pressing the contact lens onto the mouse eyeball. One single application of eye gel was sufficient in keeping the cornea moist for a 2~4 hr imaging session. During imaging, mice were stabilized on a bite-bar on a 3D translational stage with two rotational degrees of freedom (Thorlabs, PR01) and the body temperature was maintained with a heating pad (Kent Scientific, RT-0515). The mouse head was carefully aligned to make the eye perpendicular to the illumination beam, minimizing off-axis aberrations and illumination clipping by the contact lens and mouse pupil. Fluorescent dyes were injected retro-orbitally into the non-imaged eye. In wild-type mice, 40–80 µL of 5% (w/v) 2 M-Da dextran-conjugated FITC was injected for vasculature visualization. In some VLDLR-KO/Sca1-GFP mice, 30–40 µL of 5 mg/mL FITC or 5 mg/mL Evans Blue were injected. To generate bright enough fluorescent guide star for direct wavefront sensing in the weakly-fluorescent mouse line *rd1*-Thy1-GCaMP6s, 20–40 µL of 5 mg/mL Evans Blue was injected. To suppress RGC hyperactivity in *rd1*-Thy1-GCaMP6s mouse retina, we retro-orbitally injected 10 µL of 2% Lidocaine into the non-imaged eye.

All imaging parameters, including laser power at the mouse pupil, are listed in *Supplementary file 1*.

## Retina dissection

Mice were first euthanized by isoflurane overdose followed by cervical dislocation. Then the eyes were removed, and the retinas were isolated and immersed in standard oxygenated (95% $O_2$, 5% $CO_2$) artificial cerebrospinal fluid (ACSF) at room temperature and pH 7.2.

## Ex vivo two-photon structural imaging of dissected Sca1-GFP mouse retinas

A commercial twophoton fluorescence microscope (Bergamo, Thorlabs) was used to image dissected Sca1-GFP mouse retinas (*Figure 5—figure supplement 1*). Two-photon excitation at 920 nm was provided by a femto-second laser (Coherent, Chameleon Ultra II). Ex vivo images were acquired by a 16× 0.8 NA water-dipping objective lens (Nikon). Hardware controls and data acquisition were performed by ThorImage.

## Multielectrode array (MEA) recordings

Isolated ex vivo rd1-Thy1-GCaMP6s retinas were cut into three pieces. Each piece was mounted onto a 60-electrode MEA chip (60ThinMEA200/300iR0ITO, Multichannel Systems) with the inner retina facing the array, so that RGCs were in close contact with electrodes. The chip was connected to an amplifier (MEA1060, Multichannel Systems) for wide-band extracellular recording of multi-unit activity. Before the onset of recording, the retina was perfused with oxygenated ACSF at 34 °C for 30 min with a flowrate of 1 mL/min. For pharmacological blockade of actional potentials, Lidocaine (2% in saline) was applied to the bath during corresponding recordings. Washout of Lidocaine was performed by continuously perfusing oxygenated ACSF at 34 °C over the course of two hours with a flowrate of 1 mL/min.

Recorded activity from RGCs were high-pass filtered at 200 Hz, digitized at 20 kHz, and analyzed offline. Extracellular spikes were defined as transient signals with peak deflection of >3.5 standard deviations from the root mean square of background signal. Because individual electrodes can detect spikes from multiple RGCs, we utilized principal component analysis to sort unique units (Offline Sorter v3, Plexon), which accepted units having interspike intervals >1ms. Each unit was compiled into a raster plot. The analysis code for processing sorted spike data into rasters is available online (https://github.com/kevjcao/Multielectrode-array, copy archived at *Cao, 2023*).

## Cell attached recordings of alpha-RGCs in *rd1*-Thy1-GCaMP6s retinas

Isolated ex vivo rd1-Thy1-GCaMP6s retinas were mounted onto filter paper (0.45 mm nitrocellulose membranes, MF-Millipore) with an optical window with the ganglion cell layer facing up. RGCs were visualized with DODT contrast infrared optics (Luigs and Neumann) and were targeted for whole cell recording with glass electrodes (4–6 MOhm) filled with ACSF. Loose- (<1 GΩ) and tight-seal patches (>1 GΩ) were obtained under voltage clamp with the command voltage set to maintain an amplifier current of 0 pA. Input resistance and series resistance were monitored throughout recording to ensure stable recording quality and cell health.

## Ex vivo two-photon calcium imaging of *rd1* mouse retina

Two-photon calcium imaging of *rd1*-Thy1-GCaMP6s retina was carried out on a custom galvo-scanning microscope equipped with a 20×1.0 NA water immersion objective (XLUMPLFLN20XW, Olympus). Excitation at 920 nm was provided by a tunable Ti:Sapphire ultrafast laser (Chameleon Ultra, Coherent). Imaging parameters were controlled by ScanImage 3.8.1 software (http://scanimage.vidriotechnologies.com/): 256×256 pixels at 1.25 Hz (2ms per line). GCaMP6s emission was collected with a GaAsP PMT shielded by a longpass filter (ET500lp, Chroma).

Isolated retinas were cut into four-leaf clovers and transferred onto filter paper (0.45 mm nitrocellulose membranes, MF-Millipore) with the ganglion cell layer facing up. Oxygenated ACSF was then perfused over the retina at 34 °C for 30 minutes with a flowrate of 1 mL/min. An initial imaging session performed to account for potential two-photon sensitivity. Experimental imaging was performed with the laser power at the sample ≤5 mW. For pharmacological blockade of actional potentials, Lidocaine (2% in saline) was applied to the bath during corresponding recordings. Washout of Lidocaine was performed by continuously perfusing oxygenated ACSF at 34 °C over the course of two hours with a flowrate of 1 mL/min.

## Image processing and analysis

All image processing, visualization, and analysis were performed in ImageJ (*Schindelin et al., 2012*). To remove motion-induced artifacts, image registration (TurboReg and StackReg plugins) was performed.

For calcium imaging analysis, the baseline fluorescence $F_0$ was determined as the $10^{th}$ percentile of raw calcium signal F(t) during Lidocaine treatment (105 s, 320 s, and 600 s recordings for *Figure 6A, B and D*, respectively), and ΔF/F was calculated as $100 \times (F(t) - F_0)/F_0 \%$.

## Acknowledgements

We thank Tyson Kim and Bingyao Tan for helpful discussions. This work was supported by US National Institutes of Health grants U01NS103489 (QZ, WC, and NJ), U01NS118300 (QZ and NJ), RF1MH120680 (YY, WC, and NJ), F32EY029983 (KJC), R01EY024334 and P30EY003176 (RHK), NIH/EY013849 (SP, CHX, and XG).

## Additional information

### Funding

| Funder | Grant reference number | Author |
|---|---|---|
| National Institutes of Health | U01NS103489 | Na Ji |
| National Institutes of Health | U01NS118300 | Na Ji |
| National Institutes of Health | RF1MH120680 | Na Ji |
| National Institutes of Health | F32EY029983 | Kevin J Cao |
| National Institutes of Health | R01EY024334 | Richard H Kramer |
| National Institutes of Health | P30EY003176 | Richard H Kramer |
| National Institutes of Health | NIH/EY013849 | Xiaohua Gong |

The funders had no role in study design, data collection and interpretation, or the decision to submit the work for publication.

### Author contributions

Qinrong Zhang, Conceptualization, Data curation, Software, Formal analysis, Validation, Investigation, Visualization, Methodology, Writing – original draft, Writing – review and editing; Yuhan Yang, Data curation, Software, Formal analysis, Validation, Investigation, Visualization, Methodology, Writing – review and editing; Kevin J Cao, Software, Formal analysis, Investigation, Visualization, Methodology, Writing – review and editing; Wei Chen, Santosh Paidi, Software, Writing – review and editing; Chun-hong Xia, Resources, Writing – review and editing; Richard H Kramer, Xiaohua Gong, Resources, Supervision, Funding acquisition, Writing – review and editing; Na Ji, Conceptualization, Resources, Data curation, Software, Formal analysis, Supervision, Funding acquisition, Validation, Investigation, Visualization, Methodology, Writing – original draft, Project administration, Writing – review and editing

### Author ORCIDs

Qinrong Zhang ⓘD http://orcid.org/0000-0001-8893-514X
Richard H Kramer ⓘD http://orcid.org/0000-0002-8755-9389
Xiaohua Gong ⓘD http://orcid.org/0000-0003-2074-1802
Na Ji ⓘD http://orcid.org/0000-0002-5527-1663

### Ethics

All animal experiments were conducted according to the National Institutes of Health guidelines for animal research. Procedures and protocols (AUP-2020-06-13343) were approved by the Institutional Animal Care and Use Committee at the University of California, Berkeley.

Decision letter and Author response
Decision letter https://doi.org/10.7554/eLife.84853.sa1
Author response https://doi.org/10.7554/eLife.84853.sa2

## Additional files

### Supplementary files
- MDAR checklist
- Supplementary file 1. Imaging parameters for all experiments.
- Source code 1. Zemax file of the eye imaging module.

### Data availability

All data generated or analyzed in this study are included in the manuscript and supporting files. Source data files have been provided for all primary figures. The Zemax file of the eye imaging module has been provided.

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
