## [Editor Report]

The authors developed a two-photon fluorescence microscope coupled with adaptive optics (AO-2PFM), allowing in vivo imaging of the mouse retinal structure and function. This new imaging system will be important for exploring normal retinal physiology and pathological alterations in retinal disease models.

---

## [Decision Letter]

**Decision letter after peer review:**

Thank you for submitting your article "Retinal microvascular and neuronal pathologies probed in vivo by adaptive optical two-photon fluorescence microscopy" for consideration by *eLife*. Your article has been reviewed by 3 peer reviewers, and the evaluation has been overseen by a Reviewing Editor and Lu Chen as the Senior Editor. The following individual involved in the review of your submission has agreed to reveal their identity: Wei Wei (Reviewer #2).

Essential revisions:

1) The authors should either tone down the conclusion or provide further supporting evidence. For details, please see comments from Reviewers #1 and #3.

2) Please discuss the recent publication (PMID: 32410964) on in vivo ca^2+^ imaging of RGC activities using scanning laser ophthalmoscopy. It will be helpful to compare the work with the 2PFM presented in the current study and point out the pros and cons of both methods.

3) Please clarify Figure 3C and D, KxKy images.

4) Reviewer #3 provided detailed suggestions on how to improve the clarity of the manuscript. Please revise accordingly.

*Reviewer #1 (Recommendations for the authors):*

This is an excellent tech development that will advance the fields of mouse retinal physiology and pathophysiology. The main concern of this reviewer is that some claims made in the manuscript are supported by examples (albeit convincing ones) without in-depth analysis. For the sample, the authors claim that they "for the first time, enabled in vivo 2PFM visualization of synaptic structures in the mouse retina", based on one image with three arrowheads pointing to several swelling structures along a couple of processes. While the profiles are in line with potential RGC postsynaptic structure, without additional synaptic markers or functional ca^2+^ signals, it is inadequate to make such a claim. Another example is the claim of microglia observed (Line 165) without presenting evidence or description. Therefore, the authors should either tune down the conclusion or provide further supporting evidence.

Please discuss the recent publication (PMID: 32410964) on in vivo ca^2+^ imaging of RGC activities using scanning laser ophthalmoscopy. It will be helpful to compare the work with the 2PFM presented in the current study and point out the pros and cons of both methods.

*Reviewer #2 (Recommendations for the authors):*

I found this manuscript beautifully written and the figures convincing.

*Reviewer #3 (Recommendations for the authors):*

The conclusions of this paper are mostly well supported by beautiful images and convincing data analysis, but some aspects of image presentation and additional data analysis may be needed to strengthen the manuscript.

1) Figure 1C: The AO image of RGC axons is clearly much improved, but more explanation is needed on how to define "synaptic structures" other than just enriched GFP signal along the axon shaft.

And also the dotted blue line "No AO (self-normed)" need to be defined and explained in the figure legend or text.

2) Figure 3E, the middle column of IPL images: Line 185 "subcellular processes could not be visualized without aberration correction (e.g., processes in the inner plexiform layer, Figure 3E, middle column).", it seems even with AO, the subcellular process is also hard to recognize. If not, can you mark the processes in this figure?

3) Figure 4C: needs quantification in addition to representative image presentation. Maybe compare the intensity profiles along both the lateral and axial directions under different FOV/sensing areas (i to iv) of individual neurons in a, b, and c?

4) Figure 5A: The yellow asterisks are supposed to label capillary disruptions, which are not obvious to this reviewer. Is the dark area in the ring-like structure the rupture? Can you mark the exact sites of disruptions in the insets, so viewers can appreciate what the capillary defects are and where they are? Similarly, in Figure 5C inset as well.

5) Figure 5D/E: using FITC, the same color as Sca1-GFP, is not helpful to determine leakage or increased background in general.

6) Figure 5F using a different color dye is much more convincing, although classifying the star-like cell as microglial just based on the morphology is premature.

7) Figure 6: Please define the F0 (baseline brightness of individual RGCs) and how it was acquired. The measurement is not visual stimuli-evoked RGC activity, it is spontaneous RGC activity. What duration of imaging recording was used as baseline measurement? And what is the cutoff of the percentage of ∆F/F for spontaneous RGC activation? This is related to Figure 6B, RGC #3,4,6 didn't seem to have spontaneous activity (the fluctuation of the GCaMP signal looks like noise, not real activation). Better to clarify. And they are very different after the Lidocaine washout. Similarly, in Figure 6D, cells #3-6 didn't seem to have hyperactivity to start with.

8) Need to provide reference of the rd1-Thy1-GCaMP6s mouse line that how this line is generated and shows GCaMP6s is only expressed in RGCs.

---

## [Author Response]

Essential revisions:1) The authors should either tone down the conclusion or provide further supporting evidence. For details, please see comments from Reviewers #1 and #3.2) Please discuss the recent publication (PMID: 32410964) on in vivo ca^2+^ imaging of RGC activities using scanning laser ophthalmoscopy. It will be helpful to compare the work with the 2PFM presented in the current study and point out the pros and cons of both methods.3) Please clarify Figure 3C and D, KxKy images.4) Reviewer #3 provided detailed suggestions on how to improve the clarity of the manuscript. Please revise accordingly.

We thank the editor for summarizing essential revisions, we have addressed all comments from the reviewers, as detailed below in our responses to individual comments.

Reviewer #1 (Recommendations for the authors):This is an excellent tech development that will advance the fields of mouse retinal physiology and pathophysiology. The main concern of this reviewer is that some claims made in the manuscript are supported by examples (albeit convincing ones) without in-depth analysis. Therefore, the authors should either tune down the conclusion or provide further supporting evidence.

We thank the reviewer for the comments and suggestions. Please find our responses below:

For the sample, the authors claim that they "for the first time, enabled in vivo 2PFM visualization of synaptic structures in the mouse retina", based on one image with three arrowheads pointing to several swelling structures along a couple of processes. While the profiles are in line with potential RGC postsynaptic structure, without additional synaptic markers or functional ca^2+^ signals, it is inadequate to make such a claim.

We agree with the reviewer that in the mouse line Thy1-GFP line M that we imaged, where GFP was expressed cytosolically, retinal synaptic structures were not specifically labeled. Our identification of synaptic structures was therefore indeed based on morphology alone. Similar morphology-based approaches have been commonly used in spine and bouton imaging. However, we agree with the reviewer that morphology alone is insufficient evidence for the presence of synapses (also see Ref [1] for a thoughtful review on the limitation of the morphology-only approach). Therefore, following the reviewer’s suggestion, we have toned down the conclusion of our observation in the manuscript:

Abstract (line 20-21): “…, is essential for resolving putative synaptic structures and achieving …”.Results – Optimized AO-2PFM for in vivo mouse retinal imaging (line 123-124): “… enabled in vivo 2PFM visualization of varicosities resembling synaptic structures in the mouse retina …”.Figure 1 caption: “White arrowheads: putative synaptic structures.”Results – Strategy for enlarging the effective area of AO correction for 3D cellular resolution imaging (line 202-203): “… this approach succeeded in resolving varicosities …”.Discussion (line 346): “… we demonstrated here the first in vivo visualization of putative retinal synaptic structures …”.When describing the resolution of our imaging system, we have chosen to keep our original statement: “synaptic resolution” (line 66 and 204). This is because “synaptic resolution” describes the technical capability of our AO-2PFM in resolving “synapse-sized” features, which, with sub-micron lateral resolution, our system is clearly capable of resolving synapses in 2D.

Another example is the claim of microglia observed (Line 165) without presenting evidence or description.

The mouse line we imaged was VLDLR-KO/Sca1-GFP. Sca1-GFP (a.k.a., Ly6a-GFP) transgenic mice express EGFP in “all functional repopulating adult hematopoietic stem cells in the adult bone marrow, and several other hematopoietic cell types” (https://www.jax.org/strain/012643, and Ref [2]). Microglia have been genetically verified to possess a myeloid nature [3]. Therefore, it is reasonable to speculate that retinal microglia in the Sca1-GFP mice may express GFP. The morphology and dynamics of the cells are consistent with activated microglial cells (e.g., Refs [4,5]). However, given the lack of labeling specificity for microglia, we have revised the claims in our manuscript:

Abstract (line 22-23): we changed from “we characterized microvascular lesions and observed microglial migration in a proliferative vascular retinopathy model” to “we characterized microvascular lesions with sub-capillary details in a proliferative vascular retinopathy model”.Introduction (line 71-73): “In our model of proliferative vascular retinopathy, AO enabled us to, for the first time, characterize retinal vascular lesions with sub-capillary details over multiple days and observe cell migration in vivo.”Results – High-resolution in vivo identification of abnormal capillaries in a pathological mouse model (line 274-281): “Moreover, on the third day, we observed GFP-positive cells within the dye-stained retinal volume that were absent in previous two days (Figure 5G). The morphology of these cells resembled that of activated microglia [4,5]. We speculated that the leakage of EB triggered local immune response and recruited ocular immune cells to the impacted area. With the subcellular resolution provided by AO-2PFM, we were able to track dynamic changes in the processes of the same cell over time (Figure 5G, bottom).”Discussion (line 410-411): “and image cell migration at subcellular resolution”.

Please discuss the recent publication (PMID: 32410964) on in vivo ca^2+^ imaging of RGC activities using scanning laser ophthalmoscopy. It will be helpful to compare the work with the 2PFM presented in the current study and point out the pros and cons of both methods.

We thank the reviewer for bringing our attention to this paper. We have further identified additional references using SLO for calcium imaging to include in Discussion (line 391-400):

“As demonstrated recently with scanning laser ophthalmoscopy^68^ (SLO), direct visualization of RGCs provides more sensitive readout of RGC degeneration than in vivo retinal characterization techniques such as optical coherence tomography and pattern electroretinogram. SLO has been combined with calcium imaging to characterize the dynamic functional changes of RGCs in disease models^69-71^. Although the current commercial availability of SLO systems can make them more easily adoptable than our homebuilt AO-2PFM, the visible excitation light utilized in SLO activates photoreceptors, requiring additional considerations in experimental designs. Without correcting the wavefront distortion introduced by the ocular tissue to both excitation and emission light, the resolution of SLO is limited to cellular level. In contrast, the NIR excitation light utilized in 2PFM minimizes photoreceptor activation; aberration correction by AO also enables subcellular resolution to be achieved in vivo.”

Reviewer #3 (Recommendations for the authors):The conclusions of this paper are mostly well supported by beautiful images and convincing data analysis, but some aspects of image presentation and additional data analysis may be needed to strengthen the manuscript.1) Figure 1C: The AO image of RGC axons is clearly much improved, but more explanation is needed on how to define "synaptic structures" other than just enriched GFP signal along the axon shaft.And also the dotted blue line "No AO (self-normed)" need to be defined and explained in the figure legend or text.

We agree with the reviewer that in the Thy1-GFP line M mice we imaged, retinal synaptic structures were not specifically labeled. Reviewer 1 raised the same concern. We have revised the manuscript as detailed in our response to Reviewer 1.

The “self-normed” was equivalent to artificially enhance the No AO profile so that its maximum is of the same value as the AO profile, in order to emphasize the much lower contrast of No AO images. However, the lack of contrast was evident from the original trace (solid blue line) and the inclusion of the self-normalized trace does not add much new info. Therefore, we now deleted this trace from Figure 1C (and Figure 2E) to avoid possible confusion.

2) Figure 3E, the middle column of IPL images: Line 185 "subcellular processes could not be visualized without aberration correction (e.g., processes in the inner plexiform layer, Figure 3E, middle column).", it seems even with AO, the subcellular process is also hard to recognize. If not, can you mark the processes in this figure?

Structures in the Thy1-YFP-16 mouse retina are densely labeled (Table 1 of Ref [8]), which makes it difficult to see subcellular processes. In Figure 3—figure supplement 1 we show some images (single sections or MIPs of thin stacks) with sub-cellular structures. It is evident from these images that visualization of subcellular structures was substantially improved by AO, as also indicated by the signal profiles across these features.

Following the reviewer’s suggestion, we have highlighted the subcellular process of one neuron in Figure 3E with a zoomed-in view.

3) Figure 4C: needs quantification in addition to representative image presentation. Maybe compare the intensity profiles along both the lateral and axial directions under different FOV/sensing areas (i to iv) of individual neurons in a, b, and c?

We thank the reviewer for this suggestion. We have included both lateral and axial line profiles of neurons imaged in the central (area a) and edge (area b) regions to Figure 4 (Figure 4D).

4) Figure 5A: The yellow asterisks are supposed to label capillary disruptions, which are not obvious to this reviewer. Is the dark area in the ring-like structure the rupture? Can you mark the exact sites of disruptions in the insets, so viewers can appreciate what the capillary defects are and where they are? Similarly, in Figure 5C inset as well.

We thank the reviewer for this suggestion, and we have marked the exact sites (the ring-like structure) with yellow asterisks (Figure 5A and C) more clearly.

We also updated the main text to describe these disruptions more clearly (line 250-254):

“Interestingly, in the VLDLR-KO/Sca1-GFP retina, images acquired with AO revealed a disruption in the capillary endothelium labeled by GFP, where the endothelial cells lined the walls of a short capillary branch but not its end face, leading to a ring-like structure (Figure 5A and C, yellow asterisks, insets i-ii; Video 3). Such ring-like structures were not observed in the WT/Sca1-GFP retina (Figure 5B).”

5) Figure 5D/E: using FITC, the same color as Sca1-GFP, is not helpful to determine leakage or increased background in general.

We chose to use FITC in the first experiment because earlier literatures reported vascular leakage of FITC in the retina of the same mouse line (e.g., Ref [9]). Thus we included the images in Figure 5D and E to show agreement with earlier results. We agree with the reviewer that (in comment (6)) using a non-green dye is more convincing, which was why we switched to Evans Blue (680 nm emission peak, Figure 5F).

6) Figure 5F using a different color dye is much more convincing, although classifying the star-like cell as microglial just based on the morphology is premature.

We thank the reviewer for this comment. Please find our response to Review 1 – point 2.

7) Figure 6: Please define the F0 (baseline brightness of individual RGCs) and how it was acquired. The measurement is not visual stimuli-evoked RGC activity, it is spontaneous RGC activity. What duration of imaging recording was used as baseline measurement?

For all the neurons (ex vivo and in vivo) in Figure 6, the baseline F0 was determined as the 10^th^ percentile of raw calcium signal F(t) during Lidocaine treatment:

Figure 6A: after Lidocaine treatment (phase ii) for 105 seconds’ recordingFigure 6B: after Lidocaine treatment (phase ii) for 320 seconds’ recordingFigure 6D: after Lidocaine treatment (phase ii) for 600 seconds’ recording

We have included how F0 was determined in our manuscript (*Materials and methods* – Image processing and analysis, line 556-558):

“For calcium imaging analysis, the baseline fluorescence F0 was determined as the 10th percentile of raw calcium signal F(t) during Lidocaine treatment (105 s, 320 s, and 600 s recordings for Figure 6A, B, and D, respectively), and ∆F/F was calculated as 100×(F(t)- F0)/ F0%.”

And what is the cutoff of the percentage of ∆F/F for spontaneous RGC activation? This is related to Figure 6B, RGC #3,4,6 didn't seem to have spontaneous activity (the fluctuation of the GCaMP signal looks like noise, not real activation). Better to clarify. And they are very different after the Lidocaine washout. Similarly, in Figure 6D, cells #3-6 didn't seem to have hyperactivity to start with.

All the recorded activity in Figure 6 was spontaneous. We did not use a cutoff for identifying spontaneous RGC activity.

The reviewer’s comments may arise from the concern that, unlike the example neuron in the original Figure 6A, the fluorescence traces of RGC #3,4,6 in Figure 6B and RGC #3-6 in Figure 6D do not show transient ∆F/F fluctuations that are typically associated from temporally varying firing rate. In ex vivo simultaneous cell-attached and 2PFM recordings, we did see hyperactive RGCs that have much “smoother” traces associated with very high firing rates. We now include examples for both types of activity patterns in the revised Figure 6A. We added to the main text (line 316-318):

“A temporally varying firing rate led to transient fluctuations in its calcium signal (ROI 1, Figure 6A) whereas a sustained high firing rate led to heightened fluorescence brightness without obvious transients (ROI 2, Figure 6A).”

RGC #3,4,6 in Figure 6B and RGC #3-6 in Figure 6D have similar characteristics to ROI2 above. Their being hyperactive was determined by the decrease of their calcium signal magnitude upon Lidocaine administration.

Regarding the differences before Lidocaine and after recovery, we speculate that for ex vivo preparation (Figure 6B), the washout of Lidocaine was incomplete. For in vivo recording (Figure 6D), washout was not realistic, thus we simply recorded 30 and 60 minutes after Lidocaine injection, which likely caused only partial recovery of hyperactivity.

8) Need to provide reference of the rd1-Thy1-GCaMP6s mouse line that how this line is generated and shows GCaMP6s is only expressed in RGCs.

We have included how this line was generated in our manuscript (Materials and methods – in vivo imaging, line 483-485):

“The *rd1*-Thy1-GCaMP6s mice were generated by crossing *rd1* (C3H/HeJ, the Jackson laboratory) with Thy1-GCaMP6s (C57BL/6J-Tg(Thy1-GCaMP6s)GP4.3Dkim/J, the Jackson laboratory).”

The Thy1 promoter shows selective expression in the ganglion cell layer of the retina [10,11]. We have included this information in our manuscript (Results – High-resolution in vivo imaging of retinal pharmacology, line 307-308):

“The *rd1*-Thy1-GCaMP6s mice selectively express GCaMP6s in their RGC layer of the retina^63,64^”

References:

1. K. P. Berry and E. Nedivi, "Spine Dynamics: Are They All the Same?," Neuron 96(1), 43–55 (2017).

2. X. Ma, C. Robin, K. Ottersbach, and E. Dzierzak, "The Ly‐6A (Sca‐1) GFP Transgene is Expressed in all Adult Mouse Hematopoietic Stem Cells," Stem Cells 20(6), 514–521 (2002).

3. F. Ginhoux, S. Lim, G. Hoeffel, D. Low, and T. Huber, "Origin and differentiation of microglia," Front. Cell. Neurosci. 7(MAR), 1–14 (2013).

4. A. Joseph, D. Power, and J. Schallek, "Imaging the dynamics of individual processes of microglia in the living retina in vivo," Biomed. Opt. Express 12(10), 6157 (2021).

5. E. Ozaki, C. Delaney, M. Campbell, and S. L. Doyle, "Minocycline suppresses disease-associated microglia (DAM) in a model of photoreceptor cell degeneration," Exp. Eye Res. 217(January), 108953 (2022).

6. C. Cheng, J. Parreno, R. B. Nowak, S. K. Biswas, K. Wang, M. Hoshino, K. Uesugi, N. Yagi, J. A. Moncaster, W.-K. Lo, B. Pierscionek, and V. M. Fowler, "Age-related changes in eye lens biomechanics, morphology, refractive index and transparency," Aging (Albany. NY). 11(24), 12497–12531 (2019).

7. C. Tan, H. na Park, J. Light, K. Lacy, and M. Pardue, "Strain differences in mouse lens refractive indices when measured with OCT," Invest. Ophthalmol. Vis. Sci. 54(15), 1917 (2013).

8. G. Feng, R. H. Mellor, M. Bernstein, C. Keller-Peck, Q. T. Nguyen, M. Wallace, J. M. Nerbonne, J. W. Lichtman, and J. R. Sanes, "Imaging Neuronal Subsets in Transgenic Mice Expressing Multiple Spectral Variants of GFP," Neuron 28(1), 41–51 (2000).

9. C. Xia, E. Lu, J. Zeng, and X. Gong, "Deletion of LRP5 in VLDLR knockout mice inhibits retinal neovascularization.," PLoS One 8(9), e75186 (2013).

10. E. E. O’Brien, U. Greferath, and E. L. Fletcher, "The effect of photoreceptor degeneration on ganglion cell morphology," J. Comp. Neurol. 522(5), 1155–1170 (2014).

11. Q. Chen, J. Cichon, W. Wang, L. Qiu, S.-J. R. Lee, N. R. Campbell, N. DeStefino, M. J. Goard, Z. Fu, R. Yasuda, L. L. Looger, B. R. Arenkiel, W.-B. Gan, and G. Feng, "Imaging Neural Activity Using Thy1-GCaMP Transgenic Mice," Neuron 76(2), 297–308 (2012).